# Conserved heat shock factors *HvHSFA2* and *HvHSFA3* control barley heat stress memory through diverged mechanisms

Loris Pratx [1], Yuri Dakhiya[1], Ruqayyah Nissen [1], Preethi Purushotham[2], Iris Hoffie [2], Jochen Kumlehn [2], Christian Kappel [1] & Isabel Bäurle [1] ✉

Climate change requires optimizing stress responses in crops. Priming and memory of heat stress (HS) allow plants to improve their tolerance against high temperatures. Here, we investigate HS memory in cultivated barley (*Hordeum vulgare*) to assess whether the mechanisms underlying priming by and memory of HS are conserved in monocots. Mutation of barley *HvHSFA2* and *HvHSFA3* reduced HS memory. This correlated with altered transcriptional responses of heat-induced genes in the mutants after recurrent HS. Conversely, overexpression of *HvHSFA2* increases HS tolerance with no penalty on productivity. While the biological role of *HSFA2* and *HSFA3* is conserved, their mechanistic functions appear to have diverged; both factors are globally required to boost induction of HS-responsive genes after recurrent HS. In summary, barley HS memory depends on the highly conserved HvHSFA2 and HvHSFA3, however, the underlying transcriptional wiring is different. Our findings provide a tangible route to improve HS tolerance in temperate cereals.

Plants are exposed to fluctuating environmental conditions that may interfere with their growth and development, and this is termed stress. One strategy of plants to adapt to recurrent stress involves priming and stress memory. Here, a low-level stress exposure allows a plant to respond more efficiently to a recurrent stress exposure after a stress-free lag (memory) interval. Such stress memory was described for both abiotic and biotic stresses[1–5]. Barley (*Hordeum vulgare* L.) is the fourth most produced cereal in the world and has a broad range of uses in animal feeding, brewing, human consumption and biofuel production[6,7]. It is also a diploid model for temperate cereals. Despite being more resilient than other crops[8], its production is predicted to suffer a strong decline due to globally increasing temperatures in the context of climate change[9,10]. The higher recurrence of HS and water scarcity is predicted to reduce global barley yields by 3 to 17%[10].

Under HS, plants develop both an immediate response that allows them to resist the acute HS and a longer-term response that primes them for recurrent HS (HS memory). Both responses are orchestrated by heat shock transcription factors (HSFs). During acute HS, HSFA1 translocates into the nucleus and induces a suite of genes, including heat shock protein (HSP) chaperones, reactive-oxygen scavengers and further HSFs, thus protecting the cells from damage[11,12]. In *Arabidopsis thaliana*, HS priming and memory are active for at least five days, and during this period, primed plants are able to better tolerate a second HS than naive plants[13,14]. *A. thaliana* HS memory is controlled by two specialized HSFs, HSFA2 and HSFA3[14–17]. They are induced after HS, their proteins interact and bind to a subset of HS-responsive genes to sustain active transcription[14,16,18]. They also trigger the deposition of chromatin marks such as histone H3 lysine 4 hyper-methylation to prime their expression. Primed *A. thaliana* memory genes exhibit sustained expression (type I) or enhanced reinduction after recurrent HS (type II)[5,16], allowing the plant to respond more effectively to recurrent HS. Notably, HSFA2 and HSFA3 are specifically required for HS memory and are dispensable for the immediate HS response[14,15].

[1]Institute of Biochemistry and Biology, University of Potsdam, 14476 Potsdam, Germany. [2]Leibniz Institute of Plant Genetics and Crop Plant Research (IPK) Gatersleben, 06466 Seeland/OT, Gatersleben, Germany. ✉e-mail: isabel.baeurle@uni-potsdam.de

The HSF family has radiated during early land plant evolution, before the separation of dicots and monocots[19]. 23 HSFs have been identified in barley[20] and 78 in hexaploid wheat[21,22]. While several HSFs have been linked with thermotolerance in barley[23] or in wheat[24–28], the orthologs of HSFA2 and HSFA3 have not been characterized, and it remains unclear whether HS memory is present in the *Triticeae*. It is presumed that stress memory is more energy-efficient than maintaining acute defense responses, which are associated with fitness and yield penalties[29–31]. Thus, improving HS resistance by modulating HS memory is a promising strategy to generate more heat-resistant crops and preserve their yield in the absence of stress.

Here, we identify and characterize HS memory in barley. We show that *HvHSFA2* and *HvHSFA3* are crucial regulators of barley HS memory that globally boost HS-induced gene expression upon recurrent HS. Interestingly, the mechanistic function of HSFA2 and HSFA3 has diverged in barley towards a more global role. Finally, we show that boosting *HSFA2* expression is a promising approach to enhance HS tolerance in temperate cereals with no or only minimally detectable productivity losses.

## Results

### Barley shows HS priming and memory

To characterize HS memory in *H. vulgare*, seedlings were primed with a two-step acclimation (ACC) treatment 2 or 3 d before being exposed to a strong heat-stress treatment (HS) on day 7 (Fig. 1a). HS memory (maTT) was assessed by comparing the phenotypes of primed plants and HS only plants (HS), as well as plants which were primed 1 h before HS (testing for acquired thermotolerance; aTT). While none of the non-primed HS plants survived the HS treatment, all aTT-treated plants survived. Notably, all or 88% of plants primed 2 d or 3 d before HS, respectively, survived the treatment, albeit with reduced dry weight (Fig. 1b, c). Thus, our data show the existence of HS memory in barley.

### Identification and expression of memory HSFs in barley

In *A. thaliana*, HSFA2 and HSFA3 transcription factors control HS memory[14–16]. To evaluate whether their function is conserved in monocots, we identified their orthologs in barley. Twelve HvHSFA sequences were obtained by BLASTp against *A. thaliana* HSFA sequences[20] and used for phylogenetic analysis together with those of

4 dicot and 5 monocot species (Fig. 2a, Supplementary Data 1). Notably, all proteins previously annotated as HSFA2 in monocots grouped with the dicot HSFA6 and A7 sequences (Supplementary Fig. 1). Conversely, dicot HSFA2 (and HSFA9) sequences were associated with monocot HSFA7a and b sequences. The monocot HSFA7b was neither found in barley nor wheat, leaving only 7HG0717290 as a candidate HSFA2 ortholog. For HSFA3, the dicot and monocot HSFA3 sequences grouped together, and indicated 2HG0134940 as the putative barley ortholog (Supplementary Fig. 2). Within the HSF domain, HSFA2 and HSFA3 showed the highest conservation with other monocot and dicot sequences in the DNA-binding subdomain, whereas the oligomerization subdomain was more diverged (Supplementary Fig. 3).

We next analyzed the expression of the barley candidate *HSFA2* and *HSFA3* genes after ACC (Fig. 2b). The putative *HSFA2* ortholog, *7HG0717290*, displayed the strongest induction after ACC (2036-fold induction immediately after ACC and 681-fold induction 2 h after ACC), in line with the strong induction of *AtHSFA2* after HS[15]. By contrast, the suggested *HSFA6* orthologs, *1HG0083950*, *4HG0402170*, *5HG0509970*, *2HG0108550*, and *5HG0519000*, exhibited a more moderate fold-induction after ACC (10, 17, 1020, 1 and 2-fold, respectively, immediately after ACC). *HvHSFA3* was maximally induced 2 h after ACC (272-fold, Fig. 2b), consistent with the reported delayed induction kinetics of *AtHSFA3*[14]. Expression of *HvHSFA2* was unaffected in *hsfa3-1* mutants, and vice versa (see below), suggesting that, as in *A. thaliana*[14], the expression of both genes is independent (Supplementary Fig. 4). In summary, *7HG0717290* and *2HG0134940* are the most likely barley orthologs of *HSFA2* and *HSFA3*, and we thus refer to them as *HvHSFA2* and *HvHSFA3*.

### *HvHSFA2* and *HvHSFA3* are required for HS memory

To investigate the involvement of *HvHSFA2* and *HvHSFA3* in HS memory, we generated knockout lines by CRISPR-associated endonuclease-mediated gene-specific mutagenesis. For each gene, two stable mutant lines homozygous for the selected mutation and not carrying the transgene-containing T-DNA were selected: *hsfa2-1* and *hsfa2-3* for *HvHSFA2*, and *hsfa3-1* and *hsfa3-3* for *HvHSFA3* (Fig. 2c, d). Both *hsfa2-1* and *hsfa2-3* exhibited reduced dry weight compared with wild type in 2 d and 3 d memory assays (Fig. 3a–d, Supplementary Fig. 5). For *hsfa2-3*, dry weight was also reduced in the aTT assay. In

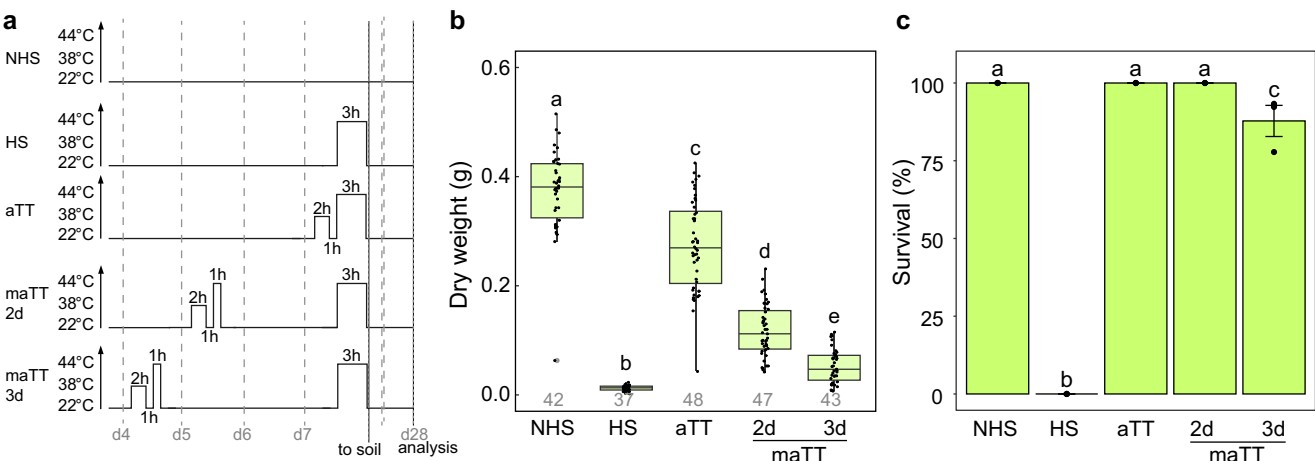

**Fig. 1 | Barley shows HS memory. a** Treatment scheme for HS assays. HS treatments were performed on 7-day-old Golden Promise plants by exposing them to 44 °C for 3 h. Plants were primed 1 h before HS by exposure to 38 °C for 2 h to test acquired thermotolerance (aTT) or 2 or 3 d before HS by exposure to an ACC treatment (38 °C for 2 h, 23 °C for 1 h and 44 °C for 1 h) to test HS memory (maTT). After HS, plants were transferred to soil and phenotyped 21 d later. **b, c** Dry weight (**b**) and survival (**c**) analysis show that priming induces tolerance to normally lethal HS. Boxplots (**b**) show median and quartile ranges with whiskers extending to 1.5 of the interquartile range of the data. Numbers below the box plots indicate the number of analyzed plants over 3 independent replicates. Data in **c** are mean ± SD from 3 independent experiments. Different letters indicate statistically significant differences ($p < 0.05$, Welch two-sided t-test (**b**), Fisher's exact test (**c**)). Source data are provided as a Source Data file.

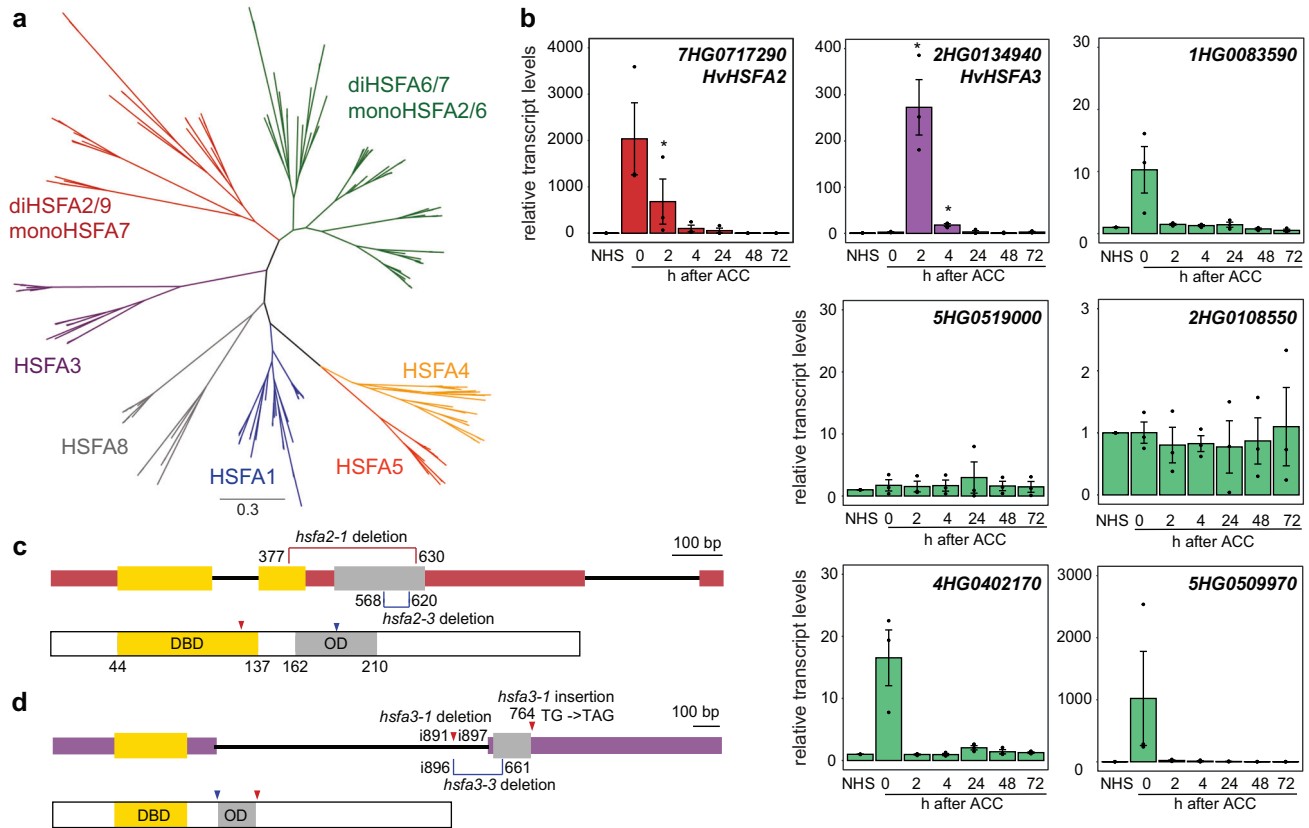

**Fig. 2 | Barley orthologs of memory HSF. a** Phylogenetic analysis of HSFA protein sequences from 4 dicot and 6 monocot species (*A. thaliana, Capsella rubella, Solanum lycopersicum, Vitis vinifera, Brachypodium distachyon, H. vulgare, Oryza sativa, Sorghum bicolor, Triticum aestivum* and *Zea mays*). HSFA2 and HSFA9 in dicots are monophyletic with HSFA7 in monocots. **b** Transcript levels of different barley *HSFA* without, or 0 to 72 h after ACC. *HvHSFA2* (*7HG0717290*) has the highest induction immediately after ACC, while *HvHSFA3* (*2HG0134940*) induction is delayed. Barley orthologs of *HSFA6/7*, including the genes previously annotated as

*HSFA2* (*2HG0108550, 5HG0519000, 5HG0509970*) exhibit no, or a smaller heat induction. Data are mean ± SEM of three independent experiments. Asterisks indicate significant differences to NHS (*, $p < 0.05$; **, $p < 0.01$; ***, $p < 0.001$; Welch two-sided t-test). Source data are provided as a Source Data file. **c, d** Schematic representation of *HvHSFA2* (**c**) and *HvHSFA3* (**d**) genes (top) and proteins (bottom). Exons are represented as squares with protein domains indicated (yellow, DBD, DNA-binding domain; gray, OD, oligomerization domain). Lesions in mutant alleles are indicated.

addition, the survival rate was lower in both mutants compared with the wild type in memory assays (Fig. 3b, d). Compared with wild type, the survival rate of *hsfa2-1* dropped from 90% to 67% and from 81% to 45% during 2 d and 3 d memory assays, respectively, and the survival rate of *hsfa2-3* dropped from 89% to 60% and from 83% to 17%. To confirm that the memory phenotype is caused by the mutation of *HvHSFA2*, we crossed *hsfa2-1* with wild type and repeated the memory assay in the segregating F2 generation. In 2 d and 3 d memory assays, plants homozygous for the *hsfa2-1* mutation had a significantly lower dry weight and a slightly lower survival compared with those that did not carry the mutation (Fig. 3e, f). Expression of a transgenic *HvHSFA2* copy driven by the constitutively expressed maize *POLYUBIQUITIN1* promoter largely complemented the memory phenotype of *hsfa2-1* (Fig. 3g, h, Supplementary Fig. 5). Any differences are likely caused by the expression strength and domain of the transgene. Thus, *HvHSFA2* is required for HS memory.

We next analyzed the *hsfa3* mutants. Both lines had reduced dry weight compared to wild type in 3 d memory assays, but not in 2 d memory assays (Fig. 3i, Supplementary Fig. 5). Dry weight was also reduced in aTT-treated *hsfa3-1*, but not *hsfa3-3* plants. Both mutants had reduced survival rates relative to wild type in 3 d memory assays (decline from 73% to 24% and 14%, respectively, Fig. 3j), and slightly reduced survival also in 2 d memory assays. To confirm that the memory phenotype is caused by the mutation of *HvHSFA3*, *hsfa3-3* was crossed with wild type, and the memory assay was repeated in a

segregating F2 family (Fig. 3k, l). Plants homozygous for *hsfa3-3* showed a significantly lower dry weight and a slightly reduced survival compared to wild type in 3 d memory assays, but not without HS treatment. Together, these results suggest that *HvHSFA3* is also required for HS memory.

We next generated and analyzed the *hsfa2-1 hsfa3-1* double mutant. In 2 d memory assays, the double mutant showed a significant decrease in survival rates compared to wild type and either single mutant (100% for wild type, 94% for *hsfa3-1*, 83% for *hsfa2-1*, 63% for *hsfa2-1 hsfa3-1*; Fig. 3m, n) and a decrease in dry weight compared to wild type and *hsfa3-1* (Fig. 3m). The stronger effect of the double mutant persisted when extending the duration of the memory period up to 5 d (Fig. 3o, p). In 5 d memory assays, the double mutant exhibited a lower dry weight compared to wild type and *hsfa3-1* and survival rates compared to wild type and either single mutant (54% for wild type, 47% for *hsfa3-1*, 36% for *hsfa2-1*, 9% for *hsfa2-1 hsfa3-1*). To complement the dry weight and survival rate measurements, we determined the relative growth rate of leaf 1 after treatments (Fig. 4a–e). In 2 d memory assays, single as well as double mutants showed reduced growth relative to wild type. In 3 d memory assays, relative growth rates were reduced in the double mutant. Notably, *hsfa2-3, hsfa3-3* and the double mutant had an increased normalized growth rate compared to wild type after both aTT and bTT treatments, which may reflect a lack of active growth repression due to inactive HS responses.

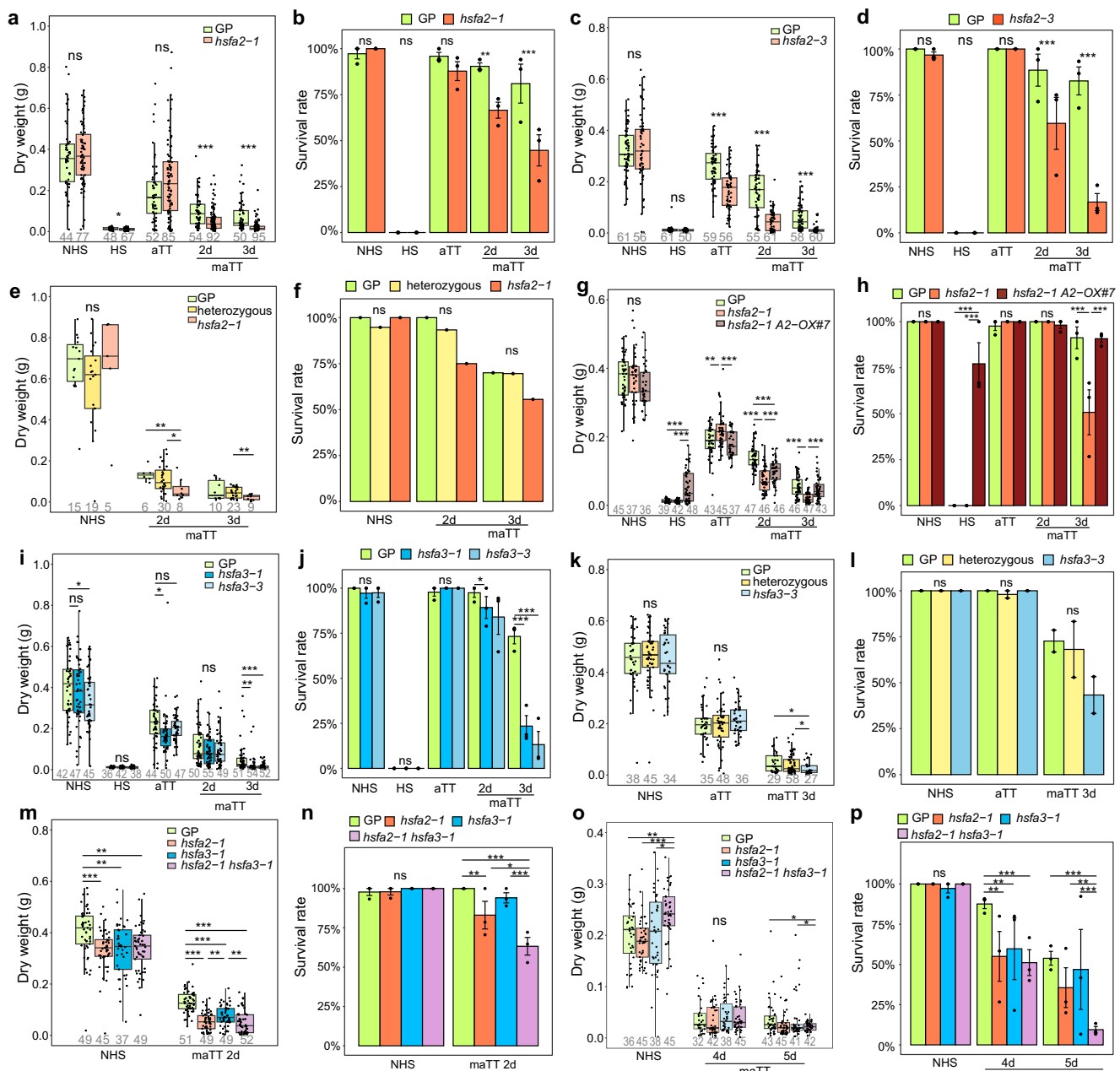

**Fig. 3 | HvHSFA2 and HvHSFA3 are required for HS memory. a–h** *HvHSFA2* is required for HS memory. **a, b** *hsfa2-1* mutants have a lower dry weight (**a**) and survival rate (**b**) than wild-type plants during memory assays, but not during acquired thermotolerance (aTT) assays, or in the absence of HS treatment. **c, d** *hsfa2-3* mutants have a lower dry weight (**c**) and survival rate (**d**) than wild type during memory and acquired thermotolerance (aTT) assays, but not in the absence of HS treatment. **e, f** The *hsfa2-1* mutation co-segregates with the mutant phenotype in a segregating F2 family from a backcross to wild type. **g, h** A transgenic copy of *HvHSFA2* complements the phenotype of *hsfa2-1*. (**i-l**) *HvHSFA3* is required for HS memory. **i, j** *hsfa3-1* and *hsfa3-3* mutants have a lower dry weight (**i**) and survival rate (**j**) during 3 d memory assays, but not during 2 d memory assays. **k, l** The *hsfa3-3* mutation co-segregates with the mutant phenotype in a segregating F2 family from a backcross to wild type. **m, n** *HvHSFA2* and *HvHSFA3* have a cumulative effect. Progeny from a segregating F2 of a cross between *hsfa2-1* and *hsfa3-1* were genotyped, and individuals homozygous for one or both of the mutations (or homozygous wild type) were phenotyped. Double mutants exhibit a lower dry weight (**m**)

and survival rate (**n**) than either of the single mutants. Among the single mutants, *hsfa2-1* has a stronger phenotype than *hsfa3-1*. **o, p** HS memory extends to 5 d. HS memory was assessed in *hsfa2-1, hsfa3-1* and in the double mutant after an extended memory period. At 5 d, the double mutant exhibited a lower dry weight (**o**) than both the wild type and the *hsfa3-1* mutant, and a lower survival (**p**) than the wild type and either single mutant. At 4 d, all mutants have a reduced survival rate. Boxplots (**a, c, e, g, i, k, m, o**) show median and quartile ranges with whiskers extending to 1.5 of the interquartile range of the data. Numbers below box plots indicate the number of analyzed plants over 3 independent experiments, except for (**e**) (one experiment) and **k** (two experiments). Data in (**b, d, f, h, j, n, p**) are the mean survival rate ± SD of 3 independent experiments, except for (**f**) (one experiment, and **l** (two experiments). Asterisks indicate significant differences (*, $p < 0.05$; **, $p < 0.01$; ***, $p < 0.001$; Welch two-sided *t*-test (**a, c, e, g, i, k, m, o**) and Fisher's exact test (**b, d, f, h, j, n, p**), respectively.). Source data are provided as a Source Data file.

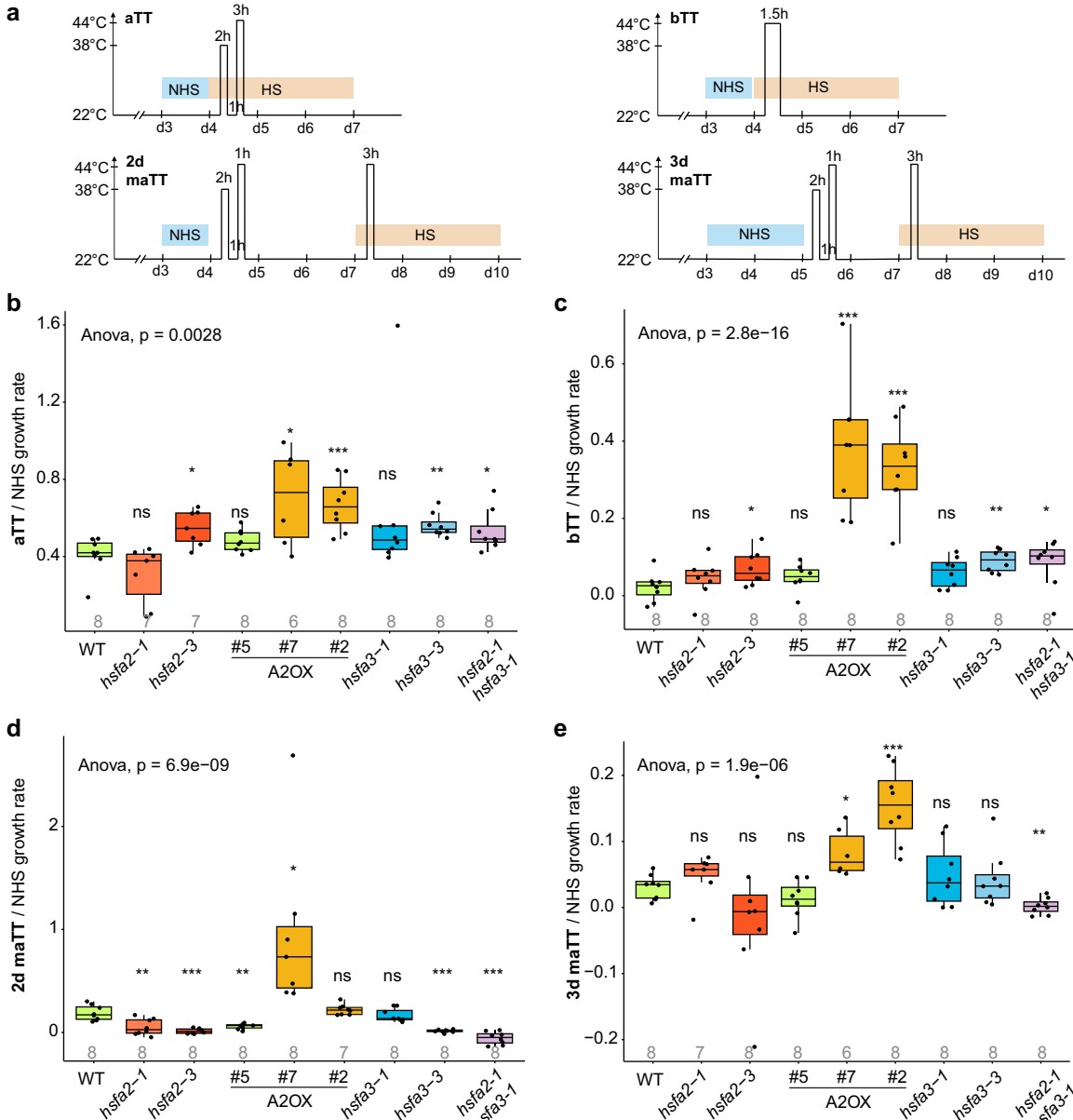

**Fig. 4 | Effects of HSFA2 and HSFA3 mutations and constitutive expression on leaf growth after HS. a** Schematic of HS treatments and measuring points. Growth rates before (NHS, blue) and after HS (orange) were determined by measuring the expansion rate of leaf 1 before and after HS treatment in the interval indicated by the colored bars. Relative growth rates after HS were calculated by normalizing the HS growth rate to the NHS growth rate for acquired thermotolerance (aTT, **b**), basal thermotolerance (bTT, **c**), or memory (maTT, **d**, **e**). Boxplots show median and quartile ranges with whiskers extending to 1.5 of the interquartile range of the data. Numbers below box plots indicate the number of replicates. Asterisks indicate significant differences compared with wild type (*, $p < 0.05$; **, $p < 0.01$; ***, $p < 0.001$; unpaired two-sided t-test). Source data are provided as a Source Data file.

In summary, our results indicate that both *HvHSFA2* and *HvHSFA3* are required for HS memory in barley, with *HvHSFA2* having a stronger role than *HvHSFA3*. Both genes have an additive effect as the double mutant performs worse than either single mutant. This is in accordance with findings from *A. thaliana*[14] and indicates that the main regulators of HS memory, as well as their relative importance, are conserved between monocot barley and dicot *A. thaliana*.

### HS-induced genes are globally misregulated in *hsfa* mutants

To investigate the molecular causes for the reduced HS memory in *hsfa2-1* and *hsfa3-1*, we analyzed transcriptomes of the mutants and wild type sampled on day 7 and treated with ACC either on day 5 (primed, P), on day 7 (triggered, T), on both days (PT) or without HS (NHS, N) (Fig. 5a). Two types of HS-induced transcriptional memory

genes were described previously; type I genes with sustained induction after HS, and type II genes with enhanced re-induction upon recurring HS[16]. Surprisingly, only four genes met the criterion of sustained HS-inducible expression for type I memory (up in P/N and T/N) (Fig. 5b, Supplementary Data 2). All of them lose their sustained induction pattern in at least one of the mutants.

We next identified type II memory genes based on their stronger re-induction after priming (log₂(PT/N and PT/T) > 1, p adj. <0.05; Fig. 5c). 82 genes fulfilled this requirement, divided into 12 +/++ (T/PT) genes and 70 0/+ genes based on their expression after T (Supplementary Data 2). None of these genes was annotated as an *HSP* gene, and no GO term was significantly enriched in this group. For the majority of these genes, the stronger re-induction was lost in *hsfa2-1* (76 genes) and/or *hsfa3-1* (71 genes) (Fig. 5d). However, the fold hyper-

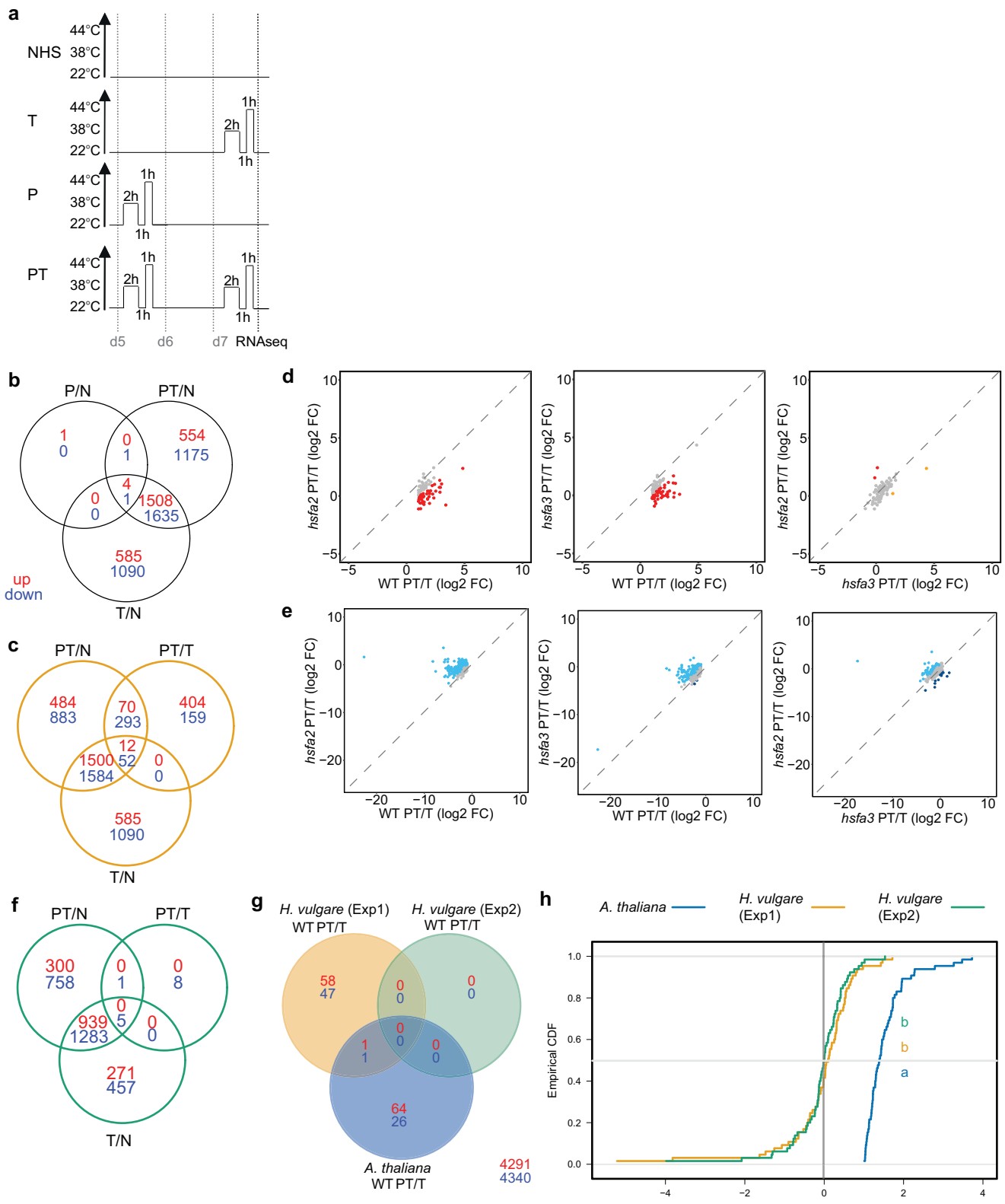

induction in the type II genes was moderate, with only 33 genes (40%) exhibiting more than three-fold hyper-induction. We confirmed the expression pattern of four type II memory genes with a high PT/T ratio and their requirement for *HvHSFA2* and *HvHSFA3* by qRT-PCR on independently grown material (Supplementary Fig. 6).

Transcriptional memory after HS and other stresses was previously associated with H3K4 hyper-methylation around the TSS[2,3,16,32–34]. We thus tested whether the confirmed memory genes accumulate H3K4me3 after priming (ACC) and 4 h, 24 or 48 h recovery (Supplementary Fig. 7). None of the tested genes showed H3K4 hyper-methylation after priming. In contrast, we detected some enrichment of H3K9ac 4 h after priming (and overall enrichment over IgG), indicating that our ChIP protocol is working. Moreover, immunoblotting for H3K4me3 and H3K9ac did not reveal a global shift between

**Fig. 5 | Analysis of genes with heat-induced Type I or Type II transcriptional memory in barley.** Transcriptomes of 7d-old plants of wild type, *hsfa2-1*, *hsfa3-1* were determined without treatment (N), immediately after ACC (triggered, T), 2 d after ACC (P) or after priming and triggering treatments (PT) by RNA-seq (**a**). Data shown in (**b**–**e**) are from Experiment 1, data shown in (**f**) are from Experiment 2, (**g**, **h**) from both. **a** Schematic of HS treatments for RNA-seq experiment. **b** Identification of type I memory genes. 4 HS-inducible (T/N) genes with sustained induction (P/N) were identified. **c** Identification of type II memory genes. 82 positive type II memory genes were identified as genes that are upregulated more strongly after a second ACC treatment (PT/N) than after a first ACC treatment (PT/T). Among them, 12 were upregulated stronger after the second treatment (+/++ genes) and 70 were upregulated after two treatments but not after one (0/+ genes). 345 negative type II memory genes were identified, among them 52 -/-- genes and 293 0/- genes. **d**, **e** Expression of Type II memory genes is impaired in *hsfa2-1*

and *hsfa3-1* mutants. Log$_2$ FC difference between PT and T in different genotypes was compared on positive (**d**) and negative (**e**) type II memory genes. In both cases, the change in expression after recurring HS was less pronounced in the mutants. **f** Identification of type II memory genes in Experiment 2 according to (**c**) identifies no positive type II memory genes and 6 negative type II memory genes (5 -/-- (T/PT) genes and 1 0/- gene). **g** Among the 197 genes with significant PT/T expression changes from *H. vulgare* or *A. thaliana* with identified orthologs in the respective other species, only two genes overlap. **h** Orthologs of *A. thaliana* memory genes with positive PT/T ratio do not show increased PT/T expression in the *H. vulgare* RNA-seq experiments. Empirical cumulative distribution function plots of PT/T ratio for *A. thaliana* PT/T up genes and their *H. vulgare* orthologs ($n = 65$). Letters on panels indicate significantly different distributions, and their colors match the colors of the genotype curves ($p < 0.05$, KS-test). Source data are provided as a Source Data file.

treatments or genotypes (Supplementary Fig. 8). We also looked for genes that showed a sustained repression after a single HS or a stronger repression after recurrent HS, analogous to the type I and type II memory genes described above. While only one gene showed sustained repression (Fig. 5b), 52 genes were repressed more after PT than after T and 293 genes were repressed only after PT but not after T (Fig. 5c). Quantitative analysis of their repression indicated that they were globally less repressed in either of the mutants (Fig. 5e).

To investigate this further, we took advantage of a second transcriptome analysis, which included the N, T, and PT treatments, in wild type and the double and single mutants of *HvHSFA2* and *HvHSFA3* (Exp. 2, Fig. 5f). Notably, this experiment did not identify any type II memory genes with the exception of six downregulated genes that were not identified in Exp. 1, confirming the lack of robust type II memory genes under our assay conditions.

To investigate whether any of the orthologs of Arabidopsis memory genes display differential expression after PT and T, we performed ortholog analyses between the barley and *A. thaliana* transcriptomes and identified 4414 orthologous gene pairs. Among these were 107 genes with differential expression between PT and T from barley Exp. 1 and 92 from *A. thaliana*. Only two of these genes overlapped (Fig. 5g). We plotted the empirical cumulative distribution function (ECDF) for the PT/T expression ratios from barley experiments 1 and 2 for the 67 barley orthologues of *A. thaliana* genes with positive PT/T (Fig. 5h). The curves from both barley experiments were centered on 0, indicating that orthologs of *A. thaliana* genes with positive PT/T as a group do not show transcriptional memory in barley. Thus, both the identity of memory genes as well as the mechanisms of priming appear to have diverged between *A. thaliana* and barley.

We thus investigated the transcriptome data more widely. In wild type, 1210 genes were upregulated after T and 1239 genes were upregulated after PT (vs. N) (Exp. 2; Fig. 6a–d, Supplementary Data 2). These 1239 genes were highly enriched for heat response and protein folding categories (Fig. 6b). Of these, 164 genes had lost the induction in *hsfa2-1* and *hsfa3-1* and the double mutant, while 676 genes were still upregulated in all three mutants (Fig. 6c). The 1210 genes upregulated after T (vs. N) behaved similarly (Fig. 6a, b).

The 1239 PT-upregulated genes were globally less induced in either of the mutants, irrespective of their fold induction in wild type ($p < 0.05$, KS-test, Fig. 6e). This suggested that *HvHSFA2* and *HvHSFA3* have a mostly quantitative effect on gene expression after HS. Indeed, analyzing all genes upregulated at PT/N in wild type as a group revealed a significantly reduced induction (ECDF curves shifted to the left) in both single and double mutants relative to wild type. This defect was more pronounced in the PT/N than T/N comparison (Fig. 6e, top two rows). When comparing the global PT/T ratio of these PT-upregulated genes, we found a small but significant reduction in all mutants (Fig. 6e, top rows, right). Notably, this reduction was stronger in the double mutant than in either single mutant, consistent with the

stronger HS memory phenotype in the double mutant (Fig. 6e, second panel, right). This effect was observed consistently in Exp. 1 and Exp. 2 (Fig. 6e, top panels, right). Together, this indicates that there is no pronounced transcriptional hyper-induction in barley after recurrent HS (PT). Rather, *HvHSFA2* and *HvHSFA3* are essential to reach full induction levels after recurrent HS (Fig. 6f). Together, our results suggest that while the overall function of *HvHSFA2* and *HvHSFA3* in HS memory is conserved between *A. thaliana* and barley, their target genes and molecular mechanisms have diverged.

Unexpectedly, we also noted a large number of genes that were downregulated after HS (Fig. 6a, b, d, Supplementary Data 2). In wild type, 2047 genes were downregulated in PT/N, of these, 328 genes had lost the downregulation in both mutants, and an additional 1294 genes were unresponsive in at least one mutant (Fig. 6c). These genes were enriched for transcription factors, auxin response phosphorylation, and defense responses (Fig. 6b). These 2047 genes were globally significantly less repressed both after T and PT treatments in the single and double mutants (Fig. 6e). Again, comparing the global PT/T expression ratio of these PT-downregulated genes identified a significantly weaker repression in all mutants, with their ECDF curves shifted to the right relative to wild type. This effect was observed consistently across both experiments (Fig. 6e, bottom panels, right). Thus, *HvHSFA2* and *HvHSFA3* are required for full repression of gene expression after recurrent HS (Fig. 6f). The molecular basis for this observation will require further investigation. In summary, *HvHSFA2* and *HvHSFA3* are not only essential for the global upregulation of gene expression after recurrent HS but also global downregulation (Fig. 6f).

## HSFA2 overexpression enhances HS tolerance in barley

We next tested whether the overexpression of HvHSFA2 could enhance HS tolerance by generating transgenic barley carrying a 3xHA-tagged *HvHSFA2* under the control of the constitutively expressed maize *POLYUBIQUITIN1* promoter, *HSFA2-OX*. Transgene expression was assessed in four independent homozygous lines by qRT-PCR and immunoblotting (Fig. 7a, b), and two *HSFA2-OX* lines that differed in their expression strength were selected for further analysis (ox#2 and ox#7). At the RNA and protein levels, ox#2 expressed the transgene ~10 times more than ox#7. After HS treatments, both HSFA2-OX lines were more tolerant and were able to survive conditions that are lethal for unprimed wild-type plants with a survival rate of 86–90%, respectively (Fig. 7c–g). *Ox*#7 performed better than wild type in the 3 d memory assay, with more than double the dry weight of wild type, while *ox*#2 performed similarly to the wild type. In addition, both lines exhibited a significantly higher normalized growth rate than wild type in almost all tested HS conditions (Fig. 4). Together, this suggests that constitutive *HvHSFA2* expression enhances thermotolerance in a similar way as a priming treatment, which is consistent with the idea that these lines are constitutively primed.

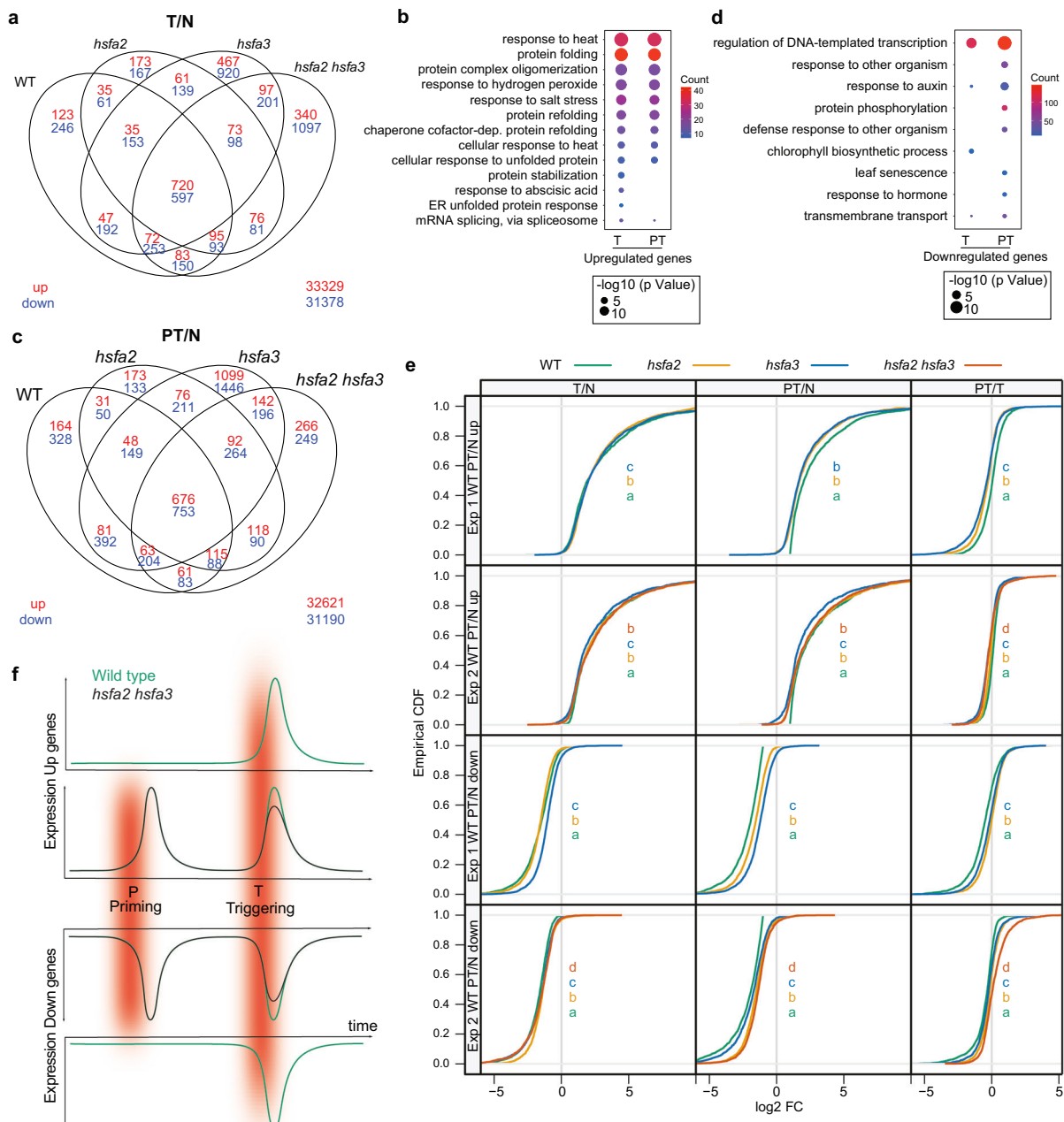

**Fig. 6 | HSFA2 and HSFA3 are required for full expression after recurrent HS (PT).** Transcriptomes of 7d-old plants of wild type, *hsfa2-1*, *hsfa3-1*, and *hsfa2 hsfa3* were determined without treatment (N), immediately after ACC (triggered, T), or after priming and triggering treatments (PT) by RNA-seq (Exp. 2). Differential expression after HS was reduced quantitatively in mutants. **a**, **c** The identification of upregulated genes after HS indicates that most of the upregulated genes after T (**a**) and PT (**c**) are shared between genotypes. **b**, **d** GO function enrichment of the genes upregulated (**b**) or downregulated (**d**) after T or PT treatment relative to N (EASE score with Benjamini-Hochberg FDR correction). **e** Empirical cumulative distribution function plots of expression changes for all genes in the groups indicated on the left (WT PT/N) up or down for Experiment 1 and Experiment 2 at the indicated timepoints and genotypes. Letters on panels indicate significantly different distributions, and their colors match the colors of the genotype curves ($p < 0.05$, KS-test). In all mutants, upregulated genes from wild type display decreased PT/T expression ratios (left-shifted ECDF curves,) and downregulated genes display increased PT/T expression ratios (right-shifted ECDF curves) in both single and double mutants compared with wild type for both experiments, consistent with HSFA2/3 being required during the PT response. **f** Model depicting the reduced expression changes in *hsfa2 and hsfa3* after PT and T relative to wild type. All panels show two curves for wild type and double mutant (color code in top panel), which overlap in the T-only panels. Source data are provided as a Source Data file.

This enhanced HS tolerance, however, was associated with a reduction in vegetative growth proportional to the expression level of *HvHSFA2*. Ox#2 and ox#7 exhibited a reduction of 69% and 43% of dry weight at 28 d relative to wild type under non-stress conditions, respectively, suggesting that constitutive expression of *HvHSFA2* interferes with normal growth (Fig. 7c, f). To estimate the productivity cost that may be associated with the enhanced expression of *HvHSFA2*, we grew ox#7 and wild-type plants to maturity in a greenhouse and determined their global productivity by assessing total grain mass and number of grains per plant (Fig. 7h, i). Interestingly, there were no significant differences between ox#7 and wild type, suggesting that moderate overexpression of *HvHSFA2* has no effects on plant

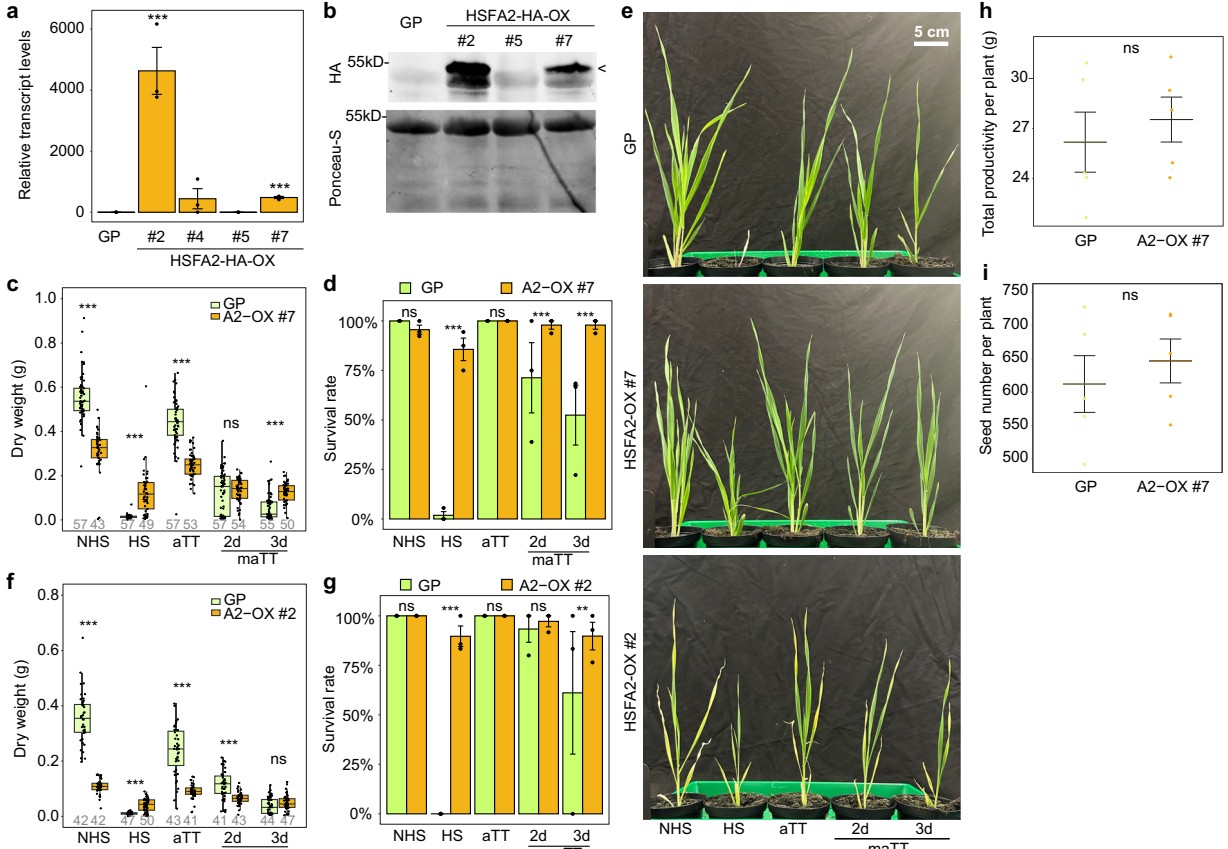

**Fig. 7 | Overexpression of *HvHSFA2* enhances HS tolerance. a, b** Expression of *pUBI::HvHSFA2-3xHA* under no HS conditions in four independent transgenic lines measured by qRT-PCR (**a**) and immunoblotting (**b**). Lines #2 and #7 were selected for further analyses. **c–g** Constitutive *HvHSFA2* expression enhances HS tolerance. Line #7 has increased dry weight (**c**) and survival rates (**d**) compared to wild type after HS and 3 d maTT treatments. Line #2 has increased dry weight (**f**) and survival rates (**g**) compared to wild type after HS and 3 d maTT (survival rate only) treatments. **e** Representative pictures of wild type and the indicated overexpressors after the indicated HS treatments (cf. Figure 1a). Plants were grown and photographed together with those in Supplementary Fig. 5. **h, i** Productivity was

estimated by determining the total grain yield (**h**) and total number of grains (**i**) per plant, displayed is the mean ± SEM of *n* = 5 plants from one experiment. Data in (**a, d, g**) are mean ± SEM of three independent experiments. The immunoblot shown in (**b**) is a representative of two independent experiments. Boxplots (**c, f**) show median and quartile ranges with whiskers extending to 1.5 of the interquartile range of the data. Numbers below box plots indicate the number of replicates from 3 independent experiments. Asterisks indicate significant differences to wild type GP (*, $p < 0.05$; **, $p < 0.01$; ***, $p < 0.001$; Welch two-sided t-test (**a, c, f, h, i**), Fisher's exact test (**d, g**)). Source data are provided as a Source Data file.

productivity, despite a lower dry mass during early development. Thus, moderate overexpression of *HvHSFA2* causes higher HS tolerance with no or minimal yield penalty. Our findings are in line with reports that *AtHSFA2* overexpression leads to an overall better thermotolerance, including basal, acquired and long-term thermotolerance[35–37].

## Discussion

HS is a major threat to global agriculture. Here we identified the barley orthologs of the memory HSFs, HSFA2 and HSFA3[14–16] and evaluated their involvement in HS memory and thermotolerance. *Hsfa2* and *hsfa3* mutants were strongly defective in HS memory. After HS, they exhibited pronounced deficits in leaf growth as well as reduced dry weight and survival rates. Mutating *hsfa2* led to a more severe phenotype than mutating *hsfa3*, and the double mutant exhibited a stronger effect than either single mutant, fully consistent with previous observations in *A. thaliana*[14].

However, in barley, the number of type I and type II HS-induced memory genes is reduced compared to *A. thaliana*[13,18,38] and their identities differ. Also, we did not detect enhanced H3K4me3 on type II memory genes, which is a hallmark of HS memory in *A. thaliana*, despite a general conservation of this mark in barley[16,33,39–41]. Thus, the mode of action of HSFA2 and HSFA3 appears to have diverged

between monocots and dicots. In barley, *HvHSFA2* and *HvHSFA3* are required globally to boost HS-inducible gene expression, particularly after recurrent HS. Thus, *HvHSFA2* and *HvHSFA3* appear to function as amplifiers of the transcriptional response, especially after recurring HS. This is reminiscent of the function of tomato *HSFA2*, which amplifies *HSP* expression after HS[42]. Interestingly, after a single HS in *A. thaliana*, AtHSFA2 and AtHSFA3 bind many early HS genes, but are dispensable for their expression[18]. While we do not know the direct target genes of HvHSFA2 and HvHSFA3, our data are consistent with a model where HS memory is achieved by HvHSFA2 and HvHSFA3 globally boosting the expression of HS-response genes after recurrent stress. Likewise, the repression of numerous downregulated genes was attenuated by the loss of HvHSFA2 and *HvHSFA3* after single and recurrent HS. It remains unclear whether these genes are direct or indirect targets of HvHSFA2/3. However, it is tempting to speculate that their repression occurs as a coordinated response to direct resources away from growth and towards stress response outputs. This is consistent with the enrichment of the GO terms DNA-dependent transcriptional regulation and auxin response in the downregulated genes. In *A. thaliana*, *HSFA2*, but not *HSFA3*, was required for Type II memory gene expression, and we were not able to pinpoint a precise molecular functional specification for either HSF in barley. While the physiological phenotype was slightly stronger in

*hsfa2* mutants, this did not have an obvious correlate in the gene expression pattern.

In view of the challenges of climate change, an important open question is whether HSFA2 and HSFA3 could be employed to improve barley HS tolerance. Increased stress tolerance is often associated with a growth penalty that might outweigh the advantages of an improved tolerance[30,31]. HSF overexpression was linked with increased HS and oxidative stress tolerance in *A. thaliana*; however, this phenotype was accompanied by growth penalties[17,36,37]. *HvHSFA2* overexpressors were more HS-tolerant, especially without priming, and tolerated HS that is lethal to a non-primed wild-type plant. Thus, constitutive expression of HSFA2 confers a "permanently primed" state to the plant, which allows a naive plant to tolerate strong HS. Our results show that moderate overexpression has no or only minimal consequences on productivity under greenhouse conditions, while still conferring high levels of thermotolerance. Thus, intermediate overexpression of *HvHSFA2* optimizes benefits and costs and may thus be suited for use in plant amelioration. It will be interesting to functionally characterize other *HvHSF* genes, such as *HSFA6*, and their role in HS tolerance and memory.

We report the identification of *HvHSFA2* and *HvHSFA3* as key regulators of HS memory in barley. Similar to their *A. thaliana* orthologs, *HvHSFA2* and *HvHSFA3* specifically control HS memory and function additively. However, their target genes and mode of gene regulation have diverged. Rather than conferring a H3K4me3-mediated transcriptional memory to their targets, *HvHSFA2* and *HvHSFA3* boost the transcriptional responses to recurring HS globally. Conversely, moderate overexpression of *HvHSFA2* generates a constitutive state of priming that strongly increases heat tolerance without loss of productivity. Thus, with fine-tuning of *HvHSFA2* expression levels and tissue specificity, the resulting constitutive priming could increase thermotolerance with no or minimal cost, and may thus be well-suited for breeding applications.

## Methods
### Plant growth conditions and treatments
*H. vulgare* (cv. Golden Promise, referred to as wild type) grains were germinated in glass containers on watered filter paper. Plants were grown under a 16/8 h light/dark cycle at 23/21 °C, seedlings were exposed to HS treatments 7 d after germination, transferred to soil and grown in a greenhouse. HS treatments were performed on 7-day-old seedlings by exposing them at 44 °C for 3 h. For aTT assays, plants were primed by exposing them to 38 °C for 120 min 1 h before HS treatment. For maTT assays, plants were primed 2 or 3 d before HS treatment with a two-step acclimation treatment (ACC, 38 °C for 2 h, 23 °C for 1 h and 44 °C for 1 h). For 4–5 d memory assays, HS was applied at day 10 instead of day 7, to allow for seedling emergence before priming.

### Plant phenotyping
Plants were phenotyped 21 d after HS treatment. Survival was assessed visually based on the presence of green tissue. Aboveground dry mass was measured by harvesting the aerial parts, drying them overnight in a drying oven, and weighing the dry tissue using a precision balance.

Additionally, a growth rate-based phenotyping method was used as a complement to the previously described phenotyping techniques. For growth rate phenotyping, plants were individually marked and imaged twice before treatment at the non-treated (NHS) phase (day 3–4, day 3–5 for 3d-maTT) and twice after treatment at the heat-shocked (HS) phase (day 4–7 for bTT and aTT, day 7–10 for 2d-maTT and 3d-maTT; Fig. 4a). The elongation rate of the first leaf was calculated from the images in both the NHS and HS phase for each plant. The ratio between growth in the HS and NHS phase was used to estimate plant fitness.

### HSFA phylogenetic analysis
Protein sequences from the indicated species, except for *H. vulgare* and *T. aestivum*, were recovered from the HEATSTER platform[19]. *H. vulgare* and *T. aestivum* sequences were identified from Morex V3[43] or IWGSC[44] genome assemblies by best reciprocal hit BLASTp against *A. thaliana* HSFA sequences. HSFAs from this dataset were classified by phylogeny. Protein sequences were aligned using MAFFT version 7.45[45] with the --auto option. Gap-rich aligned regions were removed using trimAl version 1.2[46] with the -gc 0.4 option. Maximum likelihood trees were generated with RAxML version 8.2.11[47] with the PROTGAMMAAUTO option. In all cases, the JTT (Jones-Taylor-Thornton) amino acid substitution matrix with gamma-distributed rate heterogeneity (gamma shape parameter range: alpha = 0.9–1.5) and empirical frequencies was selected as the best-fitting model. Bootstrap support was assessed using the autoMRE stopping criterion (150-450 replicates depending on the dataset). The trees were visualized with FigTree version 1.4.4[48].

### Targeted mutagenesis of *HSFA2* and *HSFA3*
Target gene-specific mutant lines were generated using the CasCADE vector system[49]. For each target gene, two genomic target motifs were selected according to published criteria[50], and the two cognate single guide RNAs were simultaneously used. Agrobacterium-mediated barley transformation was conducted according to a published protocol[51]. The sequence of mutations was determined by PCR and sequencing. Based on the predicted effect on the protein, two lines with independent mutations were selected for each gene: *hsfa2-1*, *hsfa2-3*, *hsfa3-1* and *hsfa3-3*. Primers were designed to genotype the mutation based on amplicon size differences due to a deletion (*hsfa2-1*, *hsfa2-3*, *hsfa3-3*) or as a dCAPS marker (hsfa3-1). After two generations, the phenotyping was performed on plants homozygous for the mutation of interest and lacking the transgenic T-DNA. For segregation assays, mutants were backcrossed with the wild type.

### Generation of overexpressing lines
For the generation of *HSFA2::3xHA* overexpressing line, the *HSFA2* coding sequence was amplified from genomic DNA with primers introducing a *Spe*I restriction site in 5' and a 3xHA sequence ending with an *Hind*III restriction site in 3'. *HSFA2::3xHA* was then inserted via *Spe*I and *Hind*III into the *UBI-ABM* vector containing *pUBI* in 5' and *t3SS::tNOS* in 3' (DNA Cloning Service, Hamburg, Germany; Supplementary Data 3). The full construct was inserted into the final *p6i-d35S* vector via *Pme*I and *Kpn*I and transformed as specified above. For complementation analysis, the *HSFA2-OX #7* line was crossed with *hsfa2-1*. Phenotyping was performed on F3 plants homozygous for the transgene and for *hsfa2-1*.

### Immunoblotting
Total protein was extracted from 500 mg leaves in Extraction buffer (250 mM Sucrose, 60 mM KCl, 15 mM NaCl, 5 mM MgCl$_2$, 15 mM PIPES, 0.8% (v/v) Triton X-100, Complete mini Protease inhibitor cocktail (Roche)). Extracted proteins were diluted 2 times in loading buffer (2 M urea, 75 mM Tris pH 6.5, 0.6% (w/v) SDS, 15% (v/v) glycerol, 1% (w/v) bromophenol blue, 0.75% (v/v) β-mercaptoethanol). 20 µl of protein was loaded per lane and separated on a 13% SDS-PAGE. Proteins were immunodetected with anti-HA (abcam ab9110, dilution 1:2000) followed by goat anti-rabbit antibodies (LiCOR 926-32211, dilution 1:5000). For histone blots, histones were purified from total protein extracts by precipitation with 0.4 M sulfuric acid and pelleted using 12 volumes of acetone. Pellets were resuspended in 100 µL of loading buffer, and 20 µL were loaded per lane. Immunoblotting was performed as above using anti-H3K4me3 (Abcam ab8580, dilution 1:2000), anti-H3K9ac (Abcam ab10812, dilution 1:2000), and anti H3pan (Diagenode C15200011, dilution 1:2000) antibodies followed by

goat anti-mouse Irdye 680CW (Licor 926-68070, dilution 1:5000) and goat anti-rabbit 800CW (Licor 926–32211, dilution 1:5000)

### RNA extraction, qRT–PCR and RNA-seq

RNA was extracted from pooled second leaves of 10 plants per sample per biological replicate, with three independent biological replicates per genotype and treatment. RNA extraction was performed with hot-phenol RNA extraction as described[14]. RNA samples were treated with DNase I (Thermo Fisher 18047019), and DNase was removed by performing another hot-phenol RNA extraction on the DNase-treated samples. For qRT-PCR, DNase-treated RNA was reverse-transcribed using SuperScript II (Thermo Fisher 18064014) according to the manufacturer's instructions. Quantification of cDNA was performed by qPCR (LightCycler480 Roche) using SYBR Green MasterMix (Promega) and LightCycler480 SW 1.5.1 software (Roche). Gene expression was normalized on the reference genes HORVU.MOREX.r3.3HG0265800, 6HG0574880 and 6HG0593680 (see below) by the comparative Ct method[52]. RNA was sent to BGI (Exp. 1) or Novogen (Exp. 2) for quality control, library preparation and sequencing on a DNBseq platform (Exp. 1) and on an Illumina NovaSeq X Plus platform (Exp. 2) (paired end mode, 2 × 150 bp).

### RNA-seq data analysis

RNA-seq data were mapped to the Morex_V3 genome[43] using STAR version 2.7.10a[53] with the parameter -quantMode GeneCounts enabled to obtain counts of reads mapping to each gene. Differential gene expression analysis was conducted using DESeq2[54]. Genes with $\log_2$ fold-changes greater than 1 or less than −1 and adjusted $p$-values below 0.05 were considered differentially expressed. Empirical cumulative distribution function (ECDF) plots were generated using the R/latticeExtra package. Statistical differences between curves were tested with the Kolmogorov–Smirnov test and visualized using a common letter display (CLD). Ortholog relationships with Arabidopsis thaliana were obtained from Ensembl Plants (Ensembl Plants Genes 61, Hordeum vulgare genes, MorexV3_pseudomolecules_assembly). One-to-one orthologs were used for overlap analysis with genes previously identified[38]. Raw data have been deposited at NCBI GEO under accession numbers GSE281029 and GSE307042.

### Reference gene identification

From the RNA-seq analysis, 8 genes with the least variation between conditions (minimal cumulative absolute fold change values, Exp. 1) were selected as putative housekeeping genes. In addition, *ACT2* (*5HG0533820*) and *GAPDH* (*6HG0571560*) were added to the list. The stability of expression of these genes was assessed by qRT-PCR between wild type and mutants (*hsfa2-1* and *hsfa3-1*), and between different treatments (NHS, T, PT) and evaluated using RefFinder[55]. This analysis showed that *3HG0265800*, *6HG0574880* and *6HG0593680* were the most stable genes in our setup (Supplementary Fig. 9), and thus they were used together as control genes for all analyses.

### Chromatin immunoprecipitation

Chromatin was extracted from the pooled second leaves of 10 plants per sample per biological replicate, with three independent biological replicates per genotype and treatment. Leaves were crosslinked under vacuum in PBS buffer with 1% (w/v) formaldehyde for 10 min on ice. Chromatin was extracted using the same methodology as described for *A. thaliana* seedlings[56]. Chromatin was fragmented using a Diagenode Bioruptor (25 cycles 30 s on/30 s off) and immunoprecipitated overnight at 4 °C with IgG (Thermo Fisher 026102, dilution 1:1000), anti-panH3 (Diagenode C15200011, dilution 1:1000), anti-H3K4me3 (Abcam ab8580, dilution 1:1000), or anti-H3K9ac (Abcam ab10812, dilution 1:1000) antibodies. Antibody–chromatin complexes were purified with Dynabeads (Thermo Fisher 10002D) and successively washed with low salt (150 mM NaCl, 1% (v/v) Triton X-100), high salt (500 mM NaCl, 1% (v/v) Triton X-100), LiCl buffer (250 mM LiCl, 1% (v/v) NP-40), and TE buffer. Chromatin was eluted with ChIP elution buffer (1% (w/v) SDS, 0.1 M NaHCO₃).

### Reporting summary

Further information on research design is available in the Nature Portfolio Reporting Summary linked to this article.

## Data availability

The RNA-seq data generated in this study have been deposited at NCBI GEO under accession codes GSE281029 and GSE307042. Source data are provided with this paper.

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

## Acknowledgments

We thank Joan Baltzer, Susanne Freund, and Sabine Sommerfeld for technical assistance and Christiane Schmidt and Doreen Mäker for excellent plant care. IB acknowledges funding from the European Research Council (ERC CoG 725295 CHROMADAPT to IB) and the DFG (SFB 1644/1 project no. 512328399 to IB). YD was supported by

postdoctoral fellowships from the Alexander-von-Humboldt and Minerva Foundations.

## Author contributions
L.P. and I.B. conceived and designed experiments. L.P., Y.D., R.N., P.P., I.H., J.K., C.K., and I.B. performed experiments and analyzed the data. L.P. and I.B. wrote the manuscript with contributions from all authors.

## Funding

## Competing interests
The authors declare no competing interests.
