## [Transparent Peer Review file · Nature Communications]

Conserved heat shock factors HvHSFA2 and HvHSFA3 control barley heat stress memory through diverged mechanisms

Corresponding Author: Professor Isabel Bäurlle

Version 0:

Reviewer comments:

Reviewer #1

(Remarks to the Author)

The authors have investigated heat stress (HS) memory in barley plant which is an important crop. They have compared the memory response of barley with the Arabidopsis plants which has been published previously. The authors have compared priming and memory response in monocot and dicot plants. They have defined AtHsfA2 and AtHsfA3 orthologs of barley. They made knockout lines of the barley HvHsfA2 and HvHsfA3 by CRISPER mutagenesis and showed that the memory response is reduced in the mutants. They produced overexpression lines of HvHsfA2 and HvHsfA3 and showed that overexpression of these Hsfs increased HS tolerance. Their finding is that HvHsfA2 and HvHsfA3 associated memory is not linked with H3K4me-mediated priming. They showed that HvHsfA2 and HvHsfA3 are globally required to induce HS-responsive genes. They showed that the enhanced HS tolerance achieved with the overexpression of HvHSFA2 is without loss of productivity. The authors claim that the mechanistic functions of HSFA2 and HSFA3 have diverged in barley plants. The work done by the authors is exhaustive.

Major comments:

1. The authors tested whether the confirmed memory genes accumulate H3K4me3 after priming (ACC) and 4 h, 24 or 48 h recovery. None of the tested genes in this study showed H3K4 hyper-methylation after priming. However, the authors have not stated the region from which the primers were designed to study the H3K4me3 levels in the memory genes. In the supplementary table, only one pair of primers is shown for each memory gene. There is a possibility that the enrichment for H3K4me3 in memory genes is not at the place from where the authors have designed the primer. Enrichment of H3K4me3 in barley may differ from Arabidopsis. Thus, it is necessary to choose different regions like promoter, TSS, genic region etc. of the memory genes to be certain that there was no enhanced H3K4me3 on type II memory genes.
2. Apart from H3K4me3 histone mark, H3K36me3 is also an active chromatin mark which may have a role in boosting the genetic expression of the memory gene. Including an analysis of H3K36me3 would strengthen the conclusions. Without investigating H3K36me3 levels, it cannot be stated that chromatin-based transcriptional memory was not associated with the targets of memory genes.
3. The heat stress memory study was conducted only for up to 3 days. Given that the role of HvHsfA2 and HvHsfA3 in dry weight and survival rate becomes more pronounced in the 3-d memory stress compared to the 2-d assay, it would be interesting to determine how long memory persists in barley. Extending the analysis to 5–7 d would provide valuable insights into whether the role of HvHsfA2 and HvHsfA3 in survival of plants becomes even more prominent over an extended period of heat stress memory.
4. Previous work has shown that the ortholog of AtHsfA2 is possibly the OsHsf6a in rice, a monocot (Singh et al. 2018 Plant Science). Based on the neighbor joining method, it turns out that AtHsfA2 has higher similarity with HvHsfA6 orthologs than 7HG0717290. A HvHsfA6 ortholog 5HG0509970 showed 1020-fold induction after ACC. It could be the case that the correct ortholog of AtHsfA2 is 5HG0509970 and not 7HG0717290. For this reason, authors may have failed to observe stress memory transcription linked with H3K4me3 in 7HG0717290. It may be relevant to study the H3K4me3 mark on the 5HG0509970.
5. In our view, working on one monocot plant and comparing it with one already published dicot plant will not be good data to generalize the observations between dicots and monocots. According to the authors, HvHsfA2 and HvHsfA3 function as an amplifier of the transcription response and similar observations have earlier been made with tomato, a dicot plant, indicating that differential response in barley is not specific to monocot response.

Minor comments:

1. A recent paper shows that barley has 23 Hsfs and HvHsfA2 family is shown to contain isoforms like HsfA2a, HsfA2b and HsfA2e (Mishra et al. 2016 Plant Gene; Mishra et al. 2024, Plant Cell Reports). In the present MS, authors can mention which specific HvHsfA2 form is being targeted.
2. There are previous reports that AtHsfA2 mutant is heat sensitive and over-expressing plants are more thermotolerant and AtHsfA3 over-expressing plants are more tolerant to oxidative stress (Liu and Charng 2013, Plant Physiology, Song et al. 2016, Mol Cells). The observations on the phenotype of plants made in this study are related to the earlier observations in Arabidopsis and these may be included in the discussion.
3. AtHsfA2 downstream is shown to regulate the expression of a battery of Hsps and other heat-responsive genes and one such gene is AtHsp101. Arabidopsis HsfA2 is shown to bind with a range of different HS gene promoters. For instance, binding of AtHsfA2 is shown with AtHsp101 promoter (Tiwari et al. 2020, Plant Journal). Similar observations have been made for rice where OsHsf6a (ortholog of AtHsfA2e) regulates OsHsp101 expression (Singh et al. 2022, Physiologia Plantarum). The authors may include a discussion on the expression profile of HvHsp100 proteins in their over-expression lines.
4. According to the graph of Fig. 3m, there appears to be no significant difference in dry weight between hsa2-1 and hsa2-1 hsa3-1 double mutant in 2 d memory assay. However, authors have claimed that the double mutant showed a stronger decrease in dry weight compared to wild type and single mutant. It is important to show whether there was a statistically significant difference in dry weights between hsa2-1 and the hsa2-1 hsa3-1 double mutant in the 2-day memory assay. Furthermore, in Fig. 3n, the star for the significant difference is not assigned for the comparison between GP and hsa2-1. Does this indicate that the decrease in survival rate was not significant in hsa2-1 compared to GP?
5. Fig. 3: Do the values on Y-axis in phenotype analysis represent the mean of 10-20 plants? What stage of plant was used to analyze dry weights, and which part of the plant was used to study dry weight? This may be indicated in the legend.
6. Fig. 6: it will be more convincing if the effect is also shown with the population of plants.
7. Authors have based over-expression analysis using one higher-producing and one lower-producing line of HvHsfA2. It is a routine practice to base the results on multiple lines, not single lines. Authors may mention this in the analysis.

Reviewer #2

(Remarks to the Author)

Reviewer #3

(Remarks to the Author)

In this study, Prax et al. investigated the function of HSFA2 and HSFA3 in barley and found that while their role in regulating heat stress memory is conserved between barley and Arabidopsis, the underlying molecular mechanisms differ, as they are required to globally enhance HS-responsive gene expression. The authors further generated HSFA2 overexpression lines and found that these lines confer heat stress tolerance without a yield penalty. The phenotypical and physiological data are clear and convincing, and the manuscript is well written. I enjoyed reading this work because it provided novel and deep information in the HS memory research area in plants, especially for those in monocots. However, this work would benefit from a more in-depth RNA-seq analysis to further elucidate the differences in regulatory mechanisms between monocots and dicots, as well as the respective roles of HSFA2 and HSFA3 in barley.

Figure 2. Although HSFA2 and HSFA3 are highly conserved TFs, I would like to see the full domain information for them in barley, along with a comparison to Arabidopsis. Additionally, although the automatic selection of the best model was applied here, its details should be included in the Materials and Methods section.

Figure 3. (e) Why are there only four individual dots in the hsa2-1 lines?

Line 150. The authors tested only two species; I think it would be better to rephrase this sentence as “conserved between Arabidopsis and barley.”

Below are the comments related to Fig. 5 and Supplementary Figs. S4-S6:

Figure 5. Given that HSFA2 and HSFA3 have additive effects, it would be interesting to test global RNA expression in double knockout mutants. Additionally, by comparing gene expression in single and double mutants, the authors could better elucidate the molecular functions of HSFA2 and HSFA3. For example, while in Arabidopsis, HSFA2 and HSFA3 interact with each other to regulate heat stress memory, they still have both redundant and distinct functions.

Is the induction of HSFA2 or HSFA3 dependent on the presence of the respective protein? In Arabidopsis, both HSFA2 and HSFA3 are downstream of HSFA1s, so the induction of either HSFA2 or HSFA3 is independent of the other. I'm curious if this is also the case in barley, where there is only one HSFA1.

Line 160. The supplemental figure is disordered.

(5a) Are there any Type I and Type II genes that are orthologs in Arabidopsis?

Is it possible that, rather than representing diversified mechanisms of HS memory, the overall HS memory effect is simply lower in barley? This could explain why the authors did not observe strong induction in gene expression and did not identify many Type I and Type II genes. Could the authors comment on this?

Type II gene expression in Arabidopsis is independent of AtHSFA3. However, I did not observe such patterns in Figure 5c and the qPCR data. Could the authors elaborate a bit more on the possible mechanisms or reasons?

What are the functions of the genes specifically upregulated and downregulated by HSFA2 and HSFA3 (Figure 5e & 5g)? Do they share similar functions, or do they also have distinct roles (Line 185: What are the genes that are differentially expressed in HSFA2 and HSFA3)?

Additionally, are the orthologs of these genes in Arabidopsis also regulated by HSFA2 or HSFA3? In other words, the

authors could compare the barley dataset with the Arabidopsis dataset using ortholog analysis to determine their similarities and differences, not only limited to those memory genes but also for the DEGs in *hsfa2* and *hsfa3* mutants in any given timepoint. This would strengthen the argument for functional divergence between these two species, as stated in Line 190: "... their target genes and molecular mechanisms have diverged." Currently, there is no direct evidence supporting this statement.

Supplementary Fig. S5: The authors tested several confirmed memory genes and observed no enrichment of H3K4me3. However, given its frequent enrichment downstream of transcription start sites or sometimes across the gene body, what were the locations of the primers used for ChIP-seq for each gene, and how were the enrichments calculated? I did not see this information mentioned elsewhere.

Although H3K4me3 enrichment after HS is a hallmark of memory genes, it is also a common observation following heat acclimation. What about the global changes in H3K4me3 after HS in barley? Is it possible that the degree of H3K4me3 enrichment is lower in barley, making the ChIP-seq signal less significant? Could the authors comment on this?

Line 175: I only see that two genes were enriched in H3K9ac, and the fold-change is not high.

Line 199: Could the authors elaborate further on this point: "The upregulated genes showed a stronger effect in *hsfa2*, while the downregulated genes showed a stronger effect in *hsfa3*"? What could be the possible reason for this observation?

Below are the comments related to Fig. 6:

Can overexpression of HSFA3 complement *hsfa2*, and vice versa? Or can HSFA2ox or HSFA3ox complement phenotypes in the double mutants? In other words, do they need to form a heterodimer to regulate HS memory in barley, similar to the case in Arabidopsis, or is a single copy of HSFA2 or HSFA3 sufficient? If generating mutants is time-consuming, could the authors comment on this or use the RNA-seq data to discuss this point?

Reviewer #4

(Remarks to the Author)

Pratz et al., presented a story related to the function of HvHSFA2 and HvHSFA3 in reducing HS memory in barley. They generated their mutant lines as well as the over-expression ones. Consequently, they concluded that the biological role of HSFA2 and HSFA3 is conserved, while their mechanistic functions appear to have diverged. Overall, it is a nice story and presented in a right way, however, the novelty of this story is relatively limited, as in Arabidopsis the function of HSFA2 and HSFA3 in heat stress memory was already well studied.

Major concerns:

1. the novelty of this study is relatively limited;
2. It is okay to use Dry weight and survival rate to assess the plants' heat resistance. How about fresh weight? much better?
3. Regarding the photo of barley, one single barley seems not that good. It is hard to know whether they are representative enough or not.
4. Many details was not described in the materials and methods section, such as how many plant was planted or used for phenotype/trait analysis.
5. Many figures should be improved, such as Figure 2b, what does "relative transcript level" mean? how was it calculated? Figure 1b and others, only "survival (%)" as a title of Y axis?
6. Some key references related to HSF were not cited.

Reviewer #5

(Remarks to the Author)

The manuscript by Pratz et al. describes a study on the molecular mechanisms underlying heat stress memory in barley (a monocotyledonous plant), with a particular focus on the functions of HvHSFA2 and HvHSFA3 and their similarities and differences compared to mechanisms in dicotyledonous plants. This research topic is of considerable scientific importance, particularly concerning climate change and its influence on crop stress resistance. However, mechanistic insights into HvHSFA2 and HvHSFA3 related to barley heat stress memory must be included to enhance the manuscript.

Below are my suggestions for improving the manuscript:

1. The detailed molecular mechanism of HvHSFA2 and HvHSFA3 in heat stress memory is insufficient. Using ChIP-seq to identify the direct targets of HvHSFA2 and HvHSFA3 might be a good choice.
2. The authors state that HSFA2 and HSFA3 are essential for heat stress memory. However, only the overexpression of HSFA2 has been examined regarding its phenotype. Additionally, investigating HSFA3 overexpression is necessary, as HSFA3 may also be a viable candidate for crop breeding focused on heat stress tolerance.
3. In Fig. 3, the data of two HSFA2 mutants can be combined in one panel.
4. The number of biological replicates for RNA-seq and ChIP experiments should be classified in the Methods section.
5. In Fig. 6, it would be better to switch panels c and d with panels f and g, which are in line with the phenotype in panel e.

Version 1:

Reviewer comments:

Reviewer #1

(Remarks to the Author)

The authors have conducted experiments to gain insights into the research topic of priming and memory response in a

monocot plant. The correct ortholog of AtHsfA2 could be 7HG0717290 or 5HG0509970, but this is not firmly known. We agree that working with 5HG0509970 would necessitate the generation of knockout mutants, which would be time-consuming. However, if the authors agree, they can bring up the point in the discussion that both 5HG0509970 and 7HG0717290 orthologs would be interesting to work with, and this manuscript has focused on the 7HG0717290 isoform. Regarding which specific HvHsfA2 form has been targeted (A2a, A2b, etc.), there is possibly no direct answer at this point. The authors show that in barley, HSP101 genes show induction after HS but do not depend on HSF2 and HSF3 in their expression, which is a surprising finding and deserves a mention in the manuscript. The results on transgenic lines are based on a single line, not multiple lines. However, the work carried out in this manuscript is exhaustive. The title of the manuscript is too broad and doesn't match the results presented. The authors can include specific HSF orthologs that they have focused on in the analysis and incorporate the term 'memory' into the manuscript title.

Reviewer #2

(Remarks to the Author)

Reviewer #3

(Remarks to the Author)

I thank the authors for generating new data to address my comments regarding Figures 5 and 6. These new analyses suggest that although the regulators (HSFA2 and HSFA3) are conserved, the downstream mechanisms differ substantially between the two species. This paves the way for new avenues and provides valuable resources for the heat stress memory research field. I have no further questions.

Reviewer #5

(Remarks to the Author)

I thank the authors for their efforts in revising the manuscript and addressing some of the points raised in the previous round of review. While the manuscript has been improved, I find that several concerns regarding the interpretation of key data, the robustness of the experimental metrics, and the statistical validation of their claims have not been adequately resolved. My detailed comments are outlined below.

1. It is well-documented in the model plant *Arabidopsis thaliana* that HSFA2 and, to a lesser extent, HSFA3 are critical regulators of basal thermotolerance. However, the data presented in both Figure 3 and Figure 4 of this manuscript show no significant difference in survival rates between the wild-type (GP) and the *hsfa2* and *hsfa3* mutants under basal heat stress conditions. This finding is surprising given the conserved nature of Hsf transcription factors. The authors should discuss potential reasons for this discrepancy.
2. The reliability of dry weight as a primary metric for thermotolerance in this study is questionable due to its variability. For instance, a comparison of the NHS controls in Figure 3a and Figure 3m reveals inconsistencies: in one panel, there is a significant difference in dry weight between GP and the *hsfa2* mutant, while in another, there is not. This variability suggests that the dry weight measurement may be susceptible to subtle environmental fluctuations, making it a less robust indicator of heat stress effects.
3. In Line 217, the authors state that "The 1239 PT-upregulated genes were globally less induced in either of the mutants irrespective of their fold induction in wild type (Fig. 6e)." While I acknowledge that a statistical test might support this claim globally, the visual evidence presented in Figure 6e is not entirely convincing. In some panels, particularly the top-left one, the lines representing the different genotypes are nearly overlapping, making the reported statistical significance difficult to interpret visually.
4. The statistical analysis presented in Figure 3g indicates that the transgenic expression of HvHSFA2 in the *hsfa2-1* background (A2-OX#7) only partially complements the mutant phenotype, as its performance remains statistically different from the wild-type (GP). The authors' description should be revised to reflect this with greater precision (e.g., stating it is a "partial complementation"). Furthermore, it would be valuable for the authors to briefly discuss potential reasons for this incomplete rescue, such as transgene expression levels or positional effects.
5. There is an inconsistency in the nomenclature for the heat stress treatment combined with priming. It is referred to as "PT" in the RNA-seq scheme (Supplementary Fig. S6) but as "P+T" in the context of Figure 5. For clarity, the authors should use a single, consistent term throughout the manuscript. Additionally, the experimental scheme shown in Supplementary Fig. S6 is essential for understanding the RNA-seq design. I strongly recommend moving this scheme from the supplementary materials into the main Figure 5 (e.g., as panel 5a) to improve the accessibility and comprehension of the experimental setup for the reader.
6. The ECDF plot in Figure 6f is a potentially effective way to visualize the priming phenomenon. However, its impact is significantly diminished by two key omissions. First, the figure lacks any quantitative statistical support to validate the visual shift in distributions. Second, the figure is missing essential labels for both the X-axis and Y-axis, which must be added for clarity.
7. In Line 214, the manuscript refers to a set of 164 genes. However, it is not immediately clear how this number is represented in or derived from the Venn diagram presented in Figure 6a. The authors should clarify this.

Reviewer #1 (Remarks to the Author):

The authors have investigated heat stress (HS) memory in barley plant which is an important crop. They have compared the memory response of barley with the Arabidopsis plants which has been published previously. The authors have compared priming and memory response in monocot and dicot plants. They have defined AtHsfA2 and AtHsfA3 orthologs of barley. They made knockout lines of the barley HvHsfA2 and HvHsfA3 by CRISPER mutagenesis and showed that the memory response is reduced in the mutants. They produced overexpression lines of HvHsfA2 and HvHsfA3 and showed that over-expression of these Hsfs increased HS tolerance. Their finding is that HvHsfA2 and HvHsfA3 associated memory is not linked with H3K4me-mediated priming. They showed that HvHsfA2 and HvHsfA3 are globally required to induce HS-responsive genes. They showed that the enhanced HS tolerance achieved with the overexpression of HvHsfA2 is without loss of productivity. The authors claim that the mechanistic functions of HsfA2 and HsfA3 have diverged in barley plants. The work done by the authors is exhaustive.

> We thank the reviewers for their constructive and helpful feedback.

Major comments:

1. The authors tested whether the confirmed memory genes accumulate H3K4me3 after priming (ACC) and 4 h, 24 or 48 h recovery. None of the tested genes in this study showed H3K4 hyper-methylation after priming. However, the authors have not stated the region from which the primers were designed to study the H3K4me3 levels in the memory genes. In the supplementary table, only one pair of primers is shown for each memory gene. There is a possibility that the enrichment for H3K4me3 in memory genes is not at the place from where the authors have designed the primer. Enrichment of H3K4me3 in barley may differ from Arabidopsis. Thus, it is necessary to choose different regions like promoter, TSS, genic region etc. of the memory genes to be certain that there was no enhanced H3K4me3 on type II memory genes.

> All amplicons are at the transcription start site (TSS) of the indicated genes. We have added this information in Supplementary Fig S8. Published profiles of H3K4me3 distribution in barley indicate that it peaks at the TSS for genes in all expression groups (Baker et al 2015, doi: [10.1111/tpj.12963](https://doi.org/10.1111/tpj.12963); Navratilova et al 2025 biorxiv, <https://doi.org/10.1101/2025.02.27.640517>). The results are consistent with previous findings from Arabidopsis and other eukaryotes. Thus, the TSS is the most suitable region. In previous experiments, we also studied the ATG region with the same results. As the data sets originate from different experimental series, they cannot be combined into the Figure and are not included.

This figure from Baker 2015 (doi: [10.1111/tpj.12963](https://doi.org/10.1111/tpj.12963)) shows H3K4me3 profiles according to expression percentiles (high (purple) > mid (green-blue) > low (green) > zero (red)). The TSS has the highest signal for all expressed genes.

2. Apart from H3K4me3 histone mark, H3K36me3 is also an active chromatin mark which may have a role in boosting the genetic expression of the memory gene. Including an analysis of H3K36me3 would strengthen the conclusions. Without investigating H3K36me3 levels, it cannot be stated that chromatin-based transcriptional memory was not associated with the targets of memory genes.

> We chose H3K4me3 as a candidate mark because of previous work that links it with priming and memory at the transcriptional level in HS memory in Arabidopsis and in other contexts in other organisms. To our knowledge, H3K36me3 has not been linked with transcriptional memory. Thus, while we fully agree that studying H3K36me3 in barley HS memory may be valuable, it would require considerable efforts that go beyond the scope of this work. As we cannot exclude that another chromatin modification is connected with HS memory in barley, we have rephrased l. 325.

3. The heat stress memory study was conducted only for up to 3 days. Given that the role of HvHsfA2 and HvHsfA3 in dry weight and survival rate becomes more pronounced in the 3-d memory stress compared to the 2-d assay, it would be interesting to determine how long memory persists in barley. Extending the analysis to 5–7 d would provide valuable insights into whether the role of HvHsfA2 and HvHsfA3 in survival of plants becomes even more prominent over an extended period of heat stress memory.

> We have added 4 and 5 d assays as Fig. 3o, p. Because of the later HS treatments, we were unable to extend the memory period any further, because the seedlings reached the top of the culture glasses that were used for germination and HS treatments. The data show that HS memory extends to 5 d and that HsFA2 and HsFA3 are still required.

4. Previous work has shown that the ortholog of AtHsfA2 is possibly the OsHsf6a in rice, a monocot (Singh et al. 2018 Plant Science). Based on the neighbor joining method, it turns out that AtHsfA2 has higher similarity with HvHsfA6 orthologs than 7HG0717290. A HvHsfA6 ortholog 5HG0509970 showed 1020-fold induction after ACC. It could be the case that the correct ortholog of AtHsfA2 is 5HG0509970 and not 7HG0717290. For this reason, authors may have failed to observe stress memory transcription linked with H3K4me3 in 7HG0717290. It may be relevant to study the H3K4me3 mark on the 5HG0509970.

> As explained in the text, we consider 7HG0717290 to be the functional ortholog of AtHsfA2 because of phylogenetic sequence analysis, HS-induction, and functional analysis of knockout lines. We cannot rule out that HsfA6 orthologs such as 5HG0509970 have additional functions in HS memory, however, investigating this question would require the generation of a knockout mutant, which is very time consuming. The induction of 5HG0509970 is only transient as it is seen immediately after the ACC but not 2 h later (Fig. 2b).

5. In our view, working on one monocot plant and comparing it with one already published dicot plant will not be good data to generalize the observations between dicots and monocots. According to the authors, HvHsfA2 and HvHsfA3 function as an amplifier of the transcription response and similar observations have earlier been made with tomato, a dicot plant, indicating that differential response in barley is not specific to monocot response.

> We agree with this comment and did not intend to make overly broad statements. We toned down statements (l. 16, l. 159) that could be misinterpreted. Since HsfA2 from tomato has not been implicated in HS memory, the role of HvHsfA2 and HvHsfA3 and tomato HsfA2 are distinct. (cf. Reviewer 3).

Minor comments:

1. A recent paper shows that barley has 23 Hsfs and HvHsfA2 family is shown to contain isoforms like HsfA2a, HsfA2b and HsfA2e (Mishra et al. 2016 Plant Gene; Mishra et al. 2024, Plant Cell Reports). In the present MS, authors can mention which specific HvHsfA2 form is being targeted.

> We report that HvHsfA2 is the gene previously annotated as HvHsf7B. For clarification, we include the previous barley genome annotation gene identifiers (V1) in Supplementary Data 1, which were used in Mishra et al 2020.

2. There are previous reports that AtHsfA2 mutant is heat sensitive and over-expressing plants are more thermotolerant and AtHsfA3 over-expressing plants are more tolerant to oxidative stress (Liu

and Charnng 2013, Plant Physiology, Song et al. 2016, Mol Cells). The observations on the phenotype of plants made in this study are related to the earlier observations in Arabidopsis and these may be included in the discussion.

> Liu 2013 is Ref. 37, Song 2016 has been included (Ref. 17).

3. AtHsfA2 downstream is shown to regulate the expression of a battery of Hsps and other heat-responsive genes and one such gene is AtHsp101. Arabidopsis HsfA2 is shown to bind with a range of different HS gene promoters. For instance, binding of AtHsfA2 is shown with AtHsp101 promoter (Tiwari et al. 2020, Plant Journal). Similar observations have been made for rice where OsHsf6a (ortholog of AtHsfA2e) regulates OsHsp101 expression (Singh et al. 2022, Physiologia Plantarum). The authors may include a discussion on the expression profile of HvHsp100 proteins in their over-expression lines.

> As shown below, the two barley *HSP101* genes (bottom two rows) show induction after HS (T, PT relative to N) but do not depend on HSFA2 and HSFA3 in their expression (genotypes are columns of the Figure: GP, wild type Golden Promise; A2, *hsfa2* mutant; A3, *hsfa3* mutant; A2xA3, double mutant). Work in Arabidopsis showed that *HSP101* is bound by HSFA2 and HSFA3 but does not depend on them for its expression (Kappel et al 2023).

4. According to the graph of Fig. 3m, there appears to be no significant difference in dry weight between *hsfa2-1* and *hsfa2-1 hsfa3-1* double mutant in 2 d memory assay. However, authors have claimed that the double mutant showed a stronger decrease in dry weight compared to wild type and single mutant. It is important to show whether there was a statistically significant difference in dry weights between *hsfa2-1* and the *hsfa2-1 hsfa3-1* double mutant in the 2-day memory assay. Furthermore, in Fig. 3n, the star for the significant difference is not assigned for the comparison between GP and *hsfa2-1*. Does this indicate that the decrease in survival rate was not significant in *hsfa2-1* compared to GP?

> We have carefully checked the results of the statistical analysis and they are complete and accurate. We have edited the sentence for specificity (l.141). The significant difference in survival in *hsfa2-1* relative to GP can be seen in Fig. 3b.

5. Fig. 3: Do the values on Y-axis in phenotype analysis represent the mean of 10-20 plants? What stage of plant was used to analyze dry weights, and which part of the plant was used to study dry weight? This may be indicated in the legend.

> The figure legend states the number of plants analyzed. We added information on how the dry weight was measured to the Methods (l. 346): “Plants were phenotyped 21 d after HS treatment. Survival was assessed visually based on the presence of green tissue. Aboveground dry mass was measured by harvesting the aerial parts, drying them overnight in a drying oven, and weighing the dry tissue using a precision balance.”

6. Fig. 6: it will be more convincing if the effect is also shown with the population of plants.

> All quantitative data in Fig. 6c-g are based on three separate experiments, each including 10-20 individuals per treatment and genotype. This information is included in the legend.

7. Authors have based over-expression analysis using one higher-producing and one lower-producing line of HvHsfA2. It is a routine practice to base the results on multiple lines, not single lines. Authors may mention this in the analysis.

> We studied two overexpression lines of the same construct in detail, one with a very high level of overexpressed protein and the other with a moderate protein level. The variation in the observed phenotypes is consistent with the level of protein expression.

Reviewer #2 (Remarks to the Author):

Reviewer #3 (Remarks to the Author):

In this study, Pratx et al. investigated the function of HsFA2 and HsFA3 in barley and found that while their role in regulating heat stress memory is conserved between barley and Arabidopsis, the

underlying molecular mechanisms differ, as they are required to globally enhance HS-responsive gene expression. The authors further generated HSFA2 overexpression lines and found that these lines confer heat stress tolerance without a yield penalty. The phenotypical and physiological data are clear and convincing, and the manuscript is well written. I enjoyed reading this work because it provided novel and deep information in the HS memory research area in plants, especially for those in monocots. However, this work would benefit from a more in-depth RNA-seq analysis to further elucidate the differences in regulatory mechanisms between monocots and dicots, as well as the respective roles of HSFA2 and HSFA3 in barley.

> We thank the reviewer for their constructive and helpful feedback.

Figure 2. Although HSFA2 and HSFA3 are highly conserved TFs, I would like to see the full domain information for them in barley, along with a comparison to Arabidopsis. Additionally, although the automatic selection of the best model was applied here, its details should be included in the Materials and Methods section.

> We provided domain description in Fig. 2c and d. We have added alignments in Supplementary Fig. S3. We have clarified the details of the phylogenetic analysis in the Methods (l.367): “In all cases, the JTT (Jones-Taylor-Thornton) amino acid substitution matrix with gamma-distributed rate heterogeneity (gamma shape parameter range: alpha = 0.9-1.5) and empirical frequencies was selected as best-fitting model. Bootstrap support was assessed using the autoMRE stopping criterion (150-450 replicates depending on the dataset).

Figure 3. (e) Why are there only four individual dots in the *hsfa2-1* lines?

> For this figure, we phenotyped and genotyped around 40 plants from a family segregating for *hsfa2-1* per treatment. For NHS, only 5 of the plants were homozygous mutant and so there are only 5 individual dots. For maTT treatments, we obtained 8 and 9 homozygous mutants, respectively (cf. Source Data file).

Line 150. The authors tested only two species; I think it would be better to rephrase this sentence as “conserved between Arabidopsis and barley.”

> We rephrased this sentence as suggested (l.159).

Below are the comments related to Fig. 5 and Supplementary Figs. S4-S6:

Figure 5. Given that HSFA2 and HSFA3 have additive effects, it would be interesting to test global RNA expression in double knockout mutants. Additionally, by comparing gene expression in single

and double mutants, the authors could better elucidate the molecular functions of HSFA2 and HSFA3. For example, while in Arabidopsis, HSFA2 and HSFA3 interact with each other to regulate heat stress memory, they still have both redundant and distinct functions.

> As suggested by the reviewer, we have performed a new RNA-seq experiment including wild type, the two single mutants and the double mutants under the N, T and PT conditions, shown as new Figs. 5 and 6 (Exp. 2). Thus, we now analyze two independent RNA-seq data sets with 3 biological replicates each. We have extended the analysis to include empirical cumulative distribution functions (ECDF), which plot the log₂-fold changes in expression of different sets of genes and are well suited to study the behaviour of groups of genes, in particular, they allow analysis of quantitative changes of many genes of a group. Thus, they complement analyses based on DGE lists with hard cut-offs of significance. The new analyses confirm fully our previous conclusions in both independent data sets. They show that both HSFA2 and 3 are required for full induction (and repression) at PT conditions and that the double mutant does not behave very differently from either single mutant. Thus, we did not identify a differential requirement of HSFA2 and HSFA3.

Is the induction of HSFA2 or HSFA3 dependent on the presence of the respective protein? In Arabidopsis, both HSFA2 and HSFA3 are downstream of HSFA1s, so the induction of either HSFA2 or HSFA3 is independent of the other. I'm curious if this is also the case in barley, where there is only one HSFA1.

> Using qRT-PCR and the treatment scheme from Fig. 2b, we show that HSFA2 and HSFA3 are induced independently from each other (added as new Suppl. Fig. S4). The RNA-seq data confirms this for HSFA2 (HSFA3 is not yet induced under these conditions) in *hsfa3* and the double mutant (cf. above reply to Reviewer #1, 3.).

Line 160. The supplemental figure is disordered.

> We have corrected the referencing error.

(5a) Are there any Type I and Type II genes that are orthologs in Arabidopsis?

> We thank the review for this suggestion. Among 4414 direct orthologs between the two species, there was essentially no overlap in Type II memory genes (Fig. 5f). The barley orthologs of the Arabidopsis Type II memory genes cumulatively do not show hyperinduction (Fig 5g). The five Type I memory genes did not have orthologs among Arabidopsis type I genes.

Is it possible that, rather than representing diversified mechanisms of HS memory, the overall HS memory effect is simply lower in barley? This could explain why the authors did not observe strong induction in gene expression and did not identify many Type I and Type II genes. Could the authors comment on this?

> At the whole plant level, HS memory is detectable for at least 5 days and depends on HSFA2 and HSFA3 (cf. Fig. 3). We tried to find HS memory at the gene expression level analogous to Arabidopsis but did not succeed (cf. Fig. 5). When looking into alternative mechanisms, we found that after recurrent HS (PT), HSFA2 and HSFA3 are both required for full gene induction (and repression) (Fig. 6). The genes that are up- or downregulated at PT cumulatively show reduced induction/repression; this is evident from a shift in the mutants' curve (Fig. 6e) to the left/right. However, the PT response in wild type is similar to the T response (Fig. 6f). Thus, HSFA2 and A3 are required to boost expression under recurrent HS (PT) globally rather than on a specific set of memory genes.

Type II gene expression in Arabidopsis is independent of AtHSFA3. However, I did not observe such patterns in Figure 5c and the qPCR data. Could the authors elaborate a bit more on the possible mechanisms or reasons?

> This is a difference between Arabidopsis and barley. We extended the discussion of this topic (l.305). "In *A. thaliana* HSFA2, but not HSFA3, was required for Type II memory gene expression, and we were not able to pinpoint a precise molecular functional specification for either HSF in barley. While the physiological phenotype was slightly stronger in *hsfa2* mutants, this did not have an obvious correlate in the gene expression pattern." We currently do not know what the mechanism is.

What are the functions of the genes specifically upregulated and downregulated by HSFA2 and HSFA3 (Figure 5e & 5g)? Do they share similar functions, or do they also have distinct roles (Line 185:

What are the genes that are differentially expressed in HSFA2 and HSFA3)?

Additionally, are the orthologs of these genes in Arabidopsis also regulated by HSFA2 or HSFA3? In other words, the authors could compare the barley dataset with the Arabidopsis dataset using ortholog analysis to determine their similarities and differences, not only limited to those memory genes but also for the DEGs in *hsfa2* and *hsfa3* mutants in any given timepoint. This would strengthen the argument for functional divergence between these two species, as stated in Line 190: "... their target genes and molecular mechanisms have diverged." Currently, there is no direct evidence supporting this statement.

> While we noted in barley a small quantitative difference in *hsfa2* and *hsfa3* mutants, they have the same qualitative functions. This view is reinforced by the analysis of the double mutant that shows a

similar effect size (Fig. 6). We have limited the ortholog analysis to the well-established Arabidopsis memory genes, as we feel that this is the most relevant process for our study, which is now robustly supported by the two independent RNA-seq analyses. This ortholog analysis shows that type II memory genes from Arabidopsis do not show hyperinduction at PT in barley (Fig. 5g). In the absence of CHIP-seq data, we prefer not to speculate whether there are any specific target genes.

Supplementary Fig. S5: The authors tested several confirmed memory genes and observed no enrichment of H3K4me3. However, given its frequent enrichment downstream of transcription start sites or sometimes across the gene body, what were the locations of the primers used for CHIP-seq for each gene, and how were the enrichments calculated? I did not see this information mentioned elsewhere.

> The enrichment is calculated relative to total H3. The amplicons are all at the TSS. We added this information to the revised Fig. S8. Please also see the response to reviewer #1 Comment 1.

Although H3K4me3 enrichment after HS is a hallmark of memory genes, it is also a common observation following heat acclimation. What about the global changes in H3K4me3 after HS in barley? Is it possible that the degree of H3K4me3 enrichment is lower in barley, making the CHIP-seq signal less significant? Could the authors comment on this?

> We have included an immunoblotting experiment that shows that H3K4me3 (and H3K9ac) levels do not change globally after HS or in the mutants (Fig S9). Published genome-wide profiles of H3K4me3 in barley indicate that the overall profile and enrichment is similar as in other organisms (Baker et al., 2015; Zha et al., 2021; Ost et al., 2023).

Line 175: I only see that two genes were enriched in H3K9ac, and the fold-change is not high.

> We have rephrased the sentence.

Line 199: Could the authors elaborate further on this point: “The upregulated genes showed a stronger effect in *hsfa2*, while the downregulated genes showed a stronger effect in *hsfa3*”? What could be the possible reason for this observation?

> We have revised the statement. The comparison of both RNA-seq experiments does not indicate a consistent difference in the strength of the gene expression phenotypes.

Below are the comments related to Fig. 6:

Can overexpression of *HSFA3* complement *hsfa2*, and vice versa? Or can *HSFA2ox* or *HSFA3ox*

complement phenotypes in the double mutants? In other words, do they need to form a heterodimer to regulate HS memory in barley, similar to the case in Arabidopsis, or is a single copy of HSFA2 or HSFA3 sufficient? If generating mutants is time-consuming, could the authors comment on this or use the RNA-seq data to discuss this point?

>We tried to generate lines overexpressing HSFA3, but were unable to recover a line with expression that could be used for analysis, as all lines were either infertile or had silencing of the transgene and the endogenous copy. We do not have the material to test whether A2ox can rescue the *hsfa3* mutant. As the reviewer points out, crossing and selecting materials is very time consuming. Given the strength of the single mutant, we speculate that HSFA2 cannot take over the function of HSFA3.

Reviewer #4 (Remarks to the Author):

Pratx et al., presented a story related to the function of HvHSFA2 and HvHSFA3 in reducing HS memory in barley. They generated their mutant lines as well as the over-expression ones. Consequently, they concluded that the biological role of HSFA2 and HSFA3 is conserved, while their mechanistic functions appear to have diverged. Overall, it is a nice story and presented in a right way, however, the novelty of this story is relatively limited, as in Arabidopsis the function of HSFA2 and HSFA3 in heat stress memory was already well studied.

> We thank the reviewer for their constructive and helpful feedback.

Major concerns:

1. the novelty of this study is relatively limited;

> We present the first report of HS memory in temperate cereals along with a functional characterization of two relevant transcription factors in this process. Additional mechanistic evidence is included in the revised version following comments from the other reviewers, for example RNA-seq in double mutants, ortholog analysis, extension of memory phase to 5 days, histone modification immunoblotting, cross-regulation HSFA2/3 (Figs. 3, 5, 6, S4, S9). We believe that in sum this provides substantial novel insight into a key acclimation response in crops.

2. It is okay to use Dry weight and survival rate to assess the plants' heat resistance. How about fresh weight? much better?

> Dry weight and survival rate capture complementary aspects of the phenotypes and are widely used in similar studies on heat tolerance (see for example Yuya Fukano et al. ,Sci. Adv.

(2023).DOI:10.1126/sciadv.abq3542; Huang, P., de-Bashan, L., Crocker, T. et al. Biol Fertil Soils 53, 199–208 (2017). <https://doi.org/10.1007/s00374-016-1160-2>. In particular, dry weight is preferred over fresh weight analysis by many researchers as it is less variable and more robust, because water content varies with watering schemes, time of day, transpiration rates etc.. Details of the dry weight measurements have been added to the Methods section.

3. Regarding the photo of barley, one single barley seems not that good. It is hard to know whether they are representative enough or not.

> The barley photos are representative pictures. The accompanying quantitative phenotyping complements the pictures. It is challenging to capture all analyzed plants in one picture.

4. Many details was not described in the materials and methods section, such as how many plant was planted or used for phenotype/trait analysis.

> We supply details on the number of plants used for trait analysis in the figure legends. All planted plants were included in the analysis of experiments.

5. Many figures should be improved, such as Figure 2b, what does “relative transcript level” mean? how was it calculated? Figure 1b and others, only “survival (%)” as a title of Y axis?

> The transcript levels were normalized to three reference genes as detailed in the Methods. Fig 1c, which the reviewer is referring to, shows the percentage survival after different heat treatments evaluated 21 d after HS treatment as detailed in the methods section plant phenotyping.

6. Some key references related to HSF were not cited.

> We apologize for not being able to cite all HSF-related references due to space limitations.

Reviewer #5 (Remarks to the Author):

The manuscript by Prax et al. describes a study on the molecular mechanisms underlying heat stress memory in barley (a monocotyledonous plant), with a particular focus on the functions of HvHSFA2 and HvHSFA3 and their similarities and differences compared to mechanisms in dicotyledonous plants. This research topic is of considerable scientific importance, particularly concerning climate change and its influence on crop stress resistance. However, mechanistic insights into HvHSFA2 and HvHSFA3 related to barley heat stress memory must be included to enhance the manuscript.

Below are my suggestions for improving the manuscript:

> We thank the reviewer for their constructive and helpful feedback.

1. The detailed molecular mechanism of HvHSFA2 and HvHSFA3 in heat stress memory is insufficient. Using ChIP-seq to identify the direct targets of HvHSFA2 and HvHSFA3 might be a good choice.

> We agree that it would be interesting to identify the direct target genes of HvHSFA2 and HvHSFA3. However, we only have a tagged overexpression line available for HSFA2 and we have not succeeded in producing a tagged (over)expression line for HvHSFA3 (see below 2.). Using overexpression lines for ChIP-seq is prone to produce off-target effects. From our experience with Arabidopsis HSFAs, this might be a problem especially for this family of transcription factors that has poor sequence distinction of binding sites (cf. Kappel, Genome Biol, 2023). Thus, suitable lines or specific antibodies would need to be produced before starting a ChIP-seq study, which is a time-consuming process in barley. Nevertheless, we provide additional experiments that enhance the mechanistic insight into the function both factors (e. g. RNA-seq in double mutants, ortholog analysis, extension of memory phase to 5 days, histone modification immunoblotting, cross-regulation HSFA2/3 (Figs. 3, 5, 6, S4, S9).

2. The authors state that HSFA2 and HSFA3 are essential for heat stress memory. However, only the overexpression of HSFA2 has been examined regarding its phenotype. Additionally, investigating HSFA3 overexpression is necessary, as HSFA3 may also be a viable candidate for crop breeding focused on heat stress tolerance.

> The analysis of *hsfa2* and *hsfa3* mutants shows that they are essential for heat stress memory. We have attempted to overexpress HvHSFA3 as well, however, we could not identify a fertile line with detectable A3 expression, as all 4 lines had silencing of both, the transgene and the endogenous HSFA3. This might suggest that high levels of HvHSFA3 affect plant development and fertility, but might also be due to technical issues with the transgene.

3. In Fig. 3, the data of two HSFA2 mutants can be combined in one panel.

> Due to space limitations in the controlled climate chambers, the data derive from separate experimental series and therefore cannot be combined into one panel.

4. The number of biological replicates for RNA-seq and ChIP experiments should be classified in the Methods section.

> We have added the information in the methods for RNA-seq and figure legend for Supplementary Fig. S8.

5. In Fig. 6, it would be better to switch panels c and d with panels f and g, which are in line with the phenotype in panel e.

> We thank the reviewer for this helpful comment, we have switched the panels.

REVIEWER COMMENTS

Reviewer #1 (Remarks to the Author):

The authors have conducted experiments to gain insights into the research topic of priming and memory response in a monocot plant. The correct ortholog of AtHsfA2 could be 7HG0717290 or 5HG0509970, but this is not firmly known. We agree that working with 5HG0509970 would necessitate the generation of knockout mutants, which would be time-consuming. However, if the authors agree, they can bring up the point in the discussion that both 5HG0509970 and 7HG0717290 orthologs would be interesting to work with, and this manuscript has focused on the 7HG0717290 isoform. Regarding which specific HvHsfA2 form has been targeted (A2a, A2b, etc.), there is possibly no direct answer at this point. The authors show that in barley, HSP101 genes show induction after HS but do not depend on HSFA2 and HSFA3 in their expression, which is a surprising finding and deserves a mention in the manuscript. The results on transgenic lines are based on a single line, not multiple lines. However, the work carried out in this manuscript is exhaustive. The title of the manuscript is too broad and doesn't match the results presented. The authors can include specific HSF orthologs that they have focused on in the analysis and incorporate the term 'memory' into the manuscript title.

> We have added a sentence in the discussion to suggest that further studies might look at other HvHSFs, including 5HG0509970/HSFA6.

As our analysis of HvHSFA2/3-dependent genes does not look at individual genes, we are worried that it would be confusing for the reader to single out HSP101, whose expression, as the reviewer points out correctly, does not depend on HvHSFA2/3. Work in Arabidopsis showed that *HSP101* is bound by HSFA2 and HSFA3 but does not depend on them for its expression (Kappel et al 2023), which is exactly in line with this finding.

The results on transgenic lines presented in Fig. 3, 4, 7 are based on two independent knockout/overexpression lines. The only exception is Fig. 7h, for which only one line was used due to plant growth space constraints.

The title "Conserved heat shock factors control barley heat stress memory through diverged mechanisms" is a concise summary of the presented study. The word limit for the title is 15 words, thus, we changed it to "Conserved heat shock factors HvHSFA2 and HvHSFA3 control barley heat stress memory through diverged mechanisms" for more precision.

Reviewer #2 (Remarks to the Author):

Reviewer #3 (Remarks to the Author):

I thank the authors for generating new data to address my comments regarding Figures 5 and 6. These new analyses suggest that although the regulators (HSFA2 and HSFA3) are conserved, the downstream mechanisms differ substantially between the two species. This paves the way for new avenues and provides valuable resources for the heat stress memory research field. I have no further questions.

Reviewer #5 (Remarks to the Author):

I thank the authors for their efforts in revising the manuscript and addressing some of the points raised in the previous round of review. While the manuscript has been improved, I find that several concerns regarding the interpretation of key data, the robustness of the experimental metrics, and the statistical validation of their claims have not been adequately resolved. My detailed comments are outlined below.

1. It is well-documented in the model plant *Arabidopsis thaliana* that HSFA2 and, to a lesser extent, HSFA3 are critical regulators of basal thermotolerance. However, the data presented in both Figure 3 and Figure 4 of this manuscript show no significant difference in survival rates between the wild-type (GP) and the *hsfa2* and *hsfa3* mutants under basal heat stress conditions. This finding is surprising given the conserved nature of Hsf transcription factors. The authors should discuss potential reasons for this discrepancy.

> *Arabidopsis* HSFA2 and HSFA3 are both specifically required for HS memory (maintenance of acquired thermotolerance) and NOT basal thermotolerance (which is the amount of strong HS that can be tolerated without prior priming with a moderate HS) (Charng et al, 2007; Friedrich et al, 2021). Here, we do not focus on basal thermotolerance, but rather HS memory. Basal thermotolerance is examined only in Fig. 4c, which shows that the mutants might have a slight tendency for increased basal thermotolerance, but certainly no decrease. Thus, there is no discrepancy.

2. The reliability of dry weight as a primary metric for thermotolerance in this study is questionable due to its variability. For instance, a comparison of the NHS controls in Figure 3a and Figure 3m reveals inconsistencies: in one panel, there is a significant difference in dry weight between GP and the *hsfa2* mutant, while in another, there is not. This variability suggests that the dry weight measurement may be susceptible to subtle environmental fluctuations, making it a less robust indicator of heat stress effects.

> We use the dry weight AND survival rates equally as metrics and provide statistical analysis for both. The only inconsistency that we are aware of between Fig. 3a and 3m is in the dry weight of the NHS (no-HS treatment) plants. While we take great care to control the experimental conditions, small variation due to small fluctuating changes cannot be ruled out. The number of independent experiments (3) and individual plants analyzed (30 to 60 plants per condition per genotype) are standard for this type of experiment and support our conclusions.

3. In Line 217, the authors state that "The 1239 PT-upregulated genes were globally less induced in

either of the mutants irrespective of their fold induction in wild type (Fig. 6e)." While I acknowledge that a statistical test might support this claim globally, the visual evidence presented in Figure 6e is not entirely convincing. In some panels, particularly the top-left one, the lines representing the different genotypes are nearly overlapping, making the reported statistical significance difficult to interpret visually.

> We added details on the statistical test in the text. The top left panel shows the expression profile at T/N, where there is only a small difference in the mutants. The difference is bigger in PT/N (top middle panel), which is why the lines are spaced further apart and this fits exactly with the proposed model (cf. Fig. 6f).

4. The statistical analysis presented in Figure 3g indicates that the transgenic expression of HvHSFA2 in the *hsfa2-1* background (A2-OX#7) only partially complements the mutant phenotype, as its performance remains statistically different from the wild-type (GP). The authors' description should be revised to reflect this with greater precision (e.g., stating it is a "partial complementation"). Furthermore, it would be valuable for the authors to briefly discuss potential reasons for this incomplete rescue, such as transgene expression levels or positional effects.

> This observation is correct for 3d maTT dry weight. However, at 2d maTT there is no significant difference between GP and A2-Ox#7 in dry weight and survival. At 3d maTT, there is no significant decrease in survival rate, only a slight decrease in dry weight. Thus, we conclude that the construct largely complements the mutant phenotype. In addition, the survival rate after HS is much better than GP. We have revised the text and suggest as potential explanation that the transgene is driven by the constitutively expressed maize POLYUBIQUITIN1 promoter, not the native promoter.

5. There is an inconsistency in the nomenclature for the heat stress treatment combined with priming. It is referred to as "PT" in the RNA-seq scheme (Supplementary Fig. S6) but as "P+T" in the context of Figure 5. For clarity, the authors should use a single, consistent term throughout the manuscript. Additionally, the experimental scheme shown in Supplementary Fig. S6 is essential for understanding the RNA-seq design. I strongly recommend moving this scheme from the supplementary materials into the main Figure 5 (e.g., as panel 5a) to improve the accessibility and comprehension of the experimental setup for the reader.

> We have changed any P+T labels to PT. We moved Supplementary Fig. S6 into new Fig. 5a.

6. The ECDF plot in Figure 6f is a potentially effective way to visualize the priming phenomenon. However, its impact is significantly diminished by two key omissions. First, the figure lacks any quantitative statistical support to validate the visual shift in distributions. Second, the figure is missing essential labels for both the X-axis and Y-axis, which must be added for clarity.

> Figure 6f contains a model and not experimental data (ECDF plot). For clarification, we have added "Expression" to the y axis. The x axis is labelled with "Time". If the reviewer is referring to 6e, the graph axes are labelled (log2 FC, empirical CDF) and the colored letters refer to the results of the statistical analysis as described in the legend.

7. In Line 214, the manuscript refers to a set of 164 genes. However, it is not immediately clear how this number is represented in or derived from the Venn diagram presented in Figure 6a. The authors should clarify this.

> As stated in line 214, the text refers to Fig. 6c.